# Latent State Marginalization as a Low-cost Approach for Improving Exploration

**Dinghuai Zhang**[*]**, Aaron Courville, Yoshua Bengio**
Mila, University de Montreal

**Qinqing Zheng, Amy Zhang, Ricky T. Q. Chen**
Meta AI (FAIR)

## Abstract

While the maximum entropy (MaxEnt) reinforcement learning (RL) framework—often touted for its exploration and robustness capabilities—is usually motivated from a probabilistic perspective, the use of deep probabilistic models has not gained much traction in practice due to their inherent complexity. In this work, we propose the adoption of latent variable policies within the MaxEnt framework, which we show can provably approximate any policy distribution, and additionally, naturally emerges under the use of world models with a latent belief state. We discuss why latent variable policies are difficult to train, how naïve approaches can fail, then subsequently introduce a series of improvements centered around low-cost marginalization of the latent state, allowing us to make full use of the latent state at minimal additional cost. We instantiate our method under the actor-critic framework, marginalizing both the actor and critic. The resulting algorithm, referred to as **S**tochastic **M**arginal **A**ctor-**C**ritic (SMAC), is simple yet effective. We experimentally validate our method on continuous control tasks, showing that effective marginalization can lead to better exploration and more robust training. Our implementation is open sourced at `https://github.com/zdhNarsil/Stochastic-Marginal-Actor-Critic`.

## 1 Introduction

A fundamental goal of machine learning is to develop methods capable of sequential decision making, where reinforcement learning (RL) has achieved great success in recent decades. One of the core problems in RL is exploration, the process by which an agent learns to interact with its environment. To this end, a useful paradigm is the principle of maximum entropy, which defines the optimal solution to be one with the highest amount of randomness that solves the task at hand. While the maximum entropy (MaxEnt) RL framework (Todorov, 2006; Rawlik et al., 2012) is often motivated for learning complex multi-modal[1] behaviors through a stochastic agent, algorithms that are most often used in

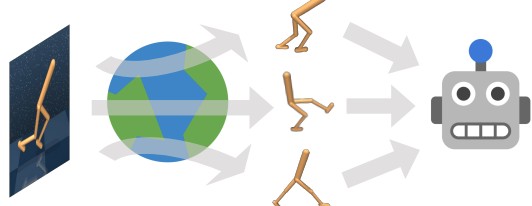

Figure 1: The world model (🌐) infers latent states (🦿) from observation inputs (🔌). While most existing methods only take one sample or the mean from this latent belief distribution, the agent (🤖) of the proposed SMAC algorithm marginalizes out the latent state for improving exploration. Icons are adapted from Mendonca et al. (2021).

practice rely on simple agents that only make local perturbations around a single action. Part of this is due to the need to compute the entropy of the agent and use it as part of the training objective.

Meanwhile, the use of more expressive models have not gained nearly as much traction in the community. While there exist works that have increased the flexibility of their agents by making use of more complex distributions such as energy-based models (Haarnoja et al., 2017), normalizing flows (Haarnoja et al., 2018a; Ward et al., 2019), mixture-of-experts (Ren et al., 2021), and

---

[*]Work done during an internship at Meta AI. Correspondence to: <`dinghuai.zhang@mila.quebec`>.

[1]By multi-modality, we're referring to distributions with different and diverse modes.

autoregressive models (Zhang et al., 2021b), these constructions often result in complicate training procedures and are inefficient in practice.

Instead, we note that a relatively simple approach to increasing expressiveness is to make use of latent variables, providing the agent with its own inference procedure for modeling stochasticity in the observations, environment, and unseen rewards. Introducing latent variables into the policy makes it possible to capture a diverse set of scenarios that are compatible with the history of observations. In particular, a majority of approaches for handling partial observability make use of world models (Hafner et al., 2019; 2020), which already result in a latent variable policy, but existing training algorithms do not make use of the latent belief state to its fullest extent. This is due in part to the fact that latent variable policies do not admit a simple expression for its entropy, and we show that naïvely estimating the entropy can lead to catastrophic failures during policy optimization. Furthermore, high-variance stochastic updates for maximizing entropy do not immediately distinguish between local random perturbations and multi-modal exploration. We propose remedies to these aforementioned downsides of latent variable policies, making use of recent advances in stochastic estimation and variance reduction. When instantiated in the actor-critic framework, the result is a simple yet effective policy optimization algorithm that can perform better exploration and lead to more robust training in both fully-observed and partially-observed settings.

Our contributions can be summarized as follows:

- We motivate the use of latent variable policies for improving exploration and robustness to partial observations, encompassing policies trained on world models as a special instance.
- We discuss the difficulties in applying latent variable policies within the MaxEnt RL paradigm. We then propose several stochastic estimation methods centered around cost-efficiency and variance reduction.
- When applied to the actor-critic framework, this yields an algorithm (SMAC; Figure 1) that is simple, effective, and adds minimal costs.
- We show through experiments that SMAC is more sample efficient and can more robustly find optimal solutions than competing actor-critic methods in both fully-observed and partially-observed continuous control tasks.

## 2 BACKGROUND

### 2.1 MAXIMUM ENTROPY REINFORCEMENT LEARNING

We first consider a standard Markov decision process (MDP) setting. We denote states $\mathbf{x}_t \in \mathcal{S}$ and actions $\mathbf{a}_t \in \mathcal{A}$, for timesteps $t \in \mathbb{N}$. There exists an initial state distribution $p(\mathbf{x}_1)$, a stochastic transition distribution $p(\mathbf{x}_t|\mathbf{x}_{t-1}, \mathbf{a}_{t-1})$, and a deterministic reward function $r_t : \mathcal{S} \times \mathcal{A} \to \mathbb{R}$. We can then learn a policy $\pi(\mathbf{a}_t|\mathbf{x}_t)$ such that the expected sum of rewards is maximized under trajectories $\tau \triangleq (\mathbf{x}_1, \mathbf{a}_1, \ldots, \mathbf{x}_T, \mathbf{a}_T)$ sampled from the policy and the transition distributions.

While it is known that the fully-observed MDP setting has at least one deterministic policy as a solution (Sutton & Barto, 2018; Puterman, 1990), efficiently searching for an optimal policy generally requires exploring sufficiently large part of the state space and keeping track of a frontier of current best solutions. As such, many works focus on the use of stochastic policies, often in conjunction with the maximum entropy (MaxEnt) framework,

$$\max_{\pi} \mathbb{E}_{p(\tau)} \left[ \sum_{t=0}^{\infty} \gamma^t \left( r_t(\mathbf{x}_t, \mathbf{a}_t) + \alpha \mathcal{H}(\pi(\cdot|\mathbf{x}_t)) \right) \right], \text{where } \mathcal{H}(\pi(\cdot|\mathbf{x}_t)) = \mathbb{E}_{\mathbf{a}_t \sim \pi(\cdot|\mathbf{x}_t)} \left[ -\log \pi(\mathbf{a}_t|\mathbf{x}_t) \right],$$

(1)

where $p(\tau)$ is the trajectory distribution with policy $\pi$, $\mathcal{H}(\cdot)$ is entropy and $\gamma$ is a discount factor.

The MaxEnt RL objective has appeared many times in the literature (*e.g.* Todorov (2006); Rawlik et al. (2012); Nachum et al. (2017)), and is recognized for its exploration (Hazan et al., 2019) and robustness (Eysenbach & Levine, 2022) capabilities. It can be equivalently interpreted as variational inference from a probabilistic modeling perspective (Norouzi et al., 2016; Levine, 2018; Lee et al., 2020a). Intuitively, MaxEnt RL encourages the policy to obtain sufficiently high reward while acting as randomly as possible, capturing the largest possible set of optimal actions. Furthermore, it also

optimizes policies to reach *future* states where it has high entropy (Haarnoja et al., 2017), resulting in improved exploration.

**Soft Actor-Critic**    A popular algorithm for solving MaxEnt RL is Soft Actor-Critic (SAC; Haarnoja et al. (2018b)), which we directly build on in this work due to its reasonable good performance and relative simplicity. Briefly, SAC alternates between learning a soft Q-function $Q(\mathbf{x}_t, \mathbf{a}_t)$ that satisfies the soft Bellman equation,

$$Q(\mathbf{x}_t, \mathbf{a}_t) = r_t(\mathbf{x}_t, \mathbf{a}_t) + \gamma \mathbb{E}_{\mathbf{a}_{t+1} \sim \pi(\cdot|\mathbf{x}_{t+1}), \mathbf{x}_{t+1} \sim p(\cdot|\mathbf{x}_t, \mathbf{a}_t)} \left[ Q(\mathbf{x}_{t+1}, \mathbf{a}_{t+1}) + \alpha \mathcal{H}(\pi(\cdot|\mathbf{x}_{t+1})) \right], \quad (2)$$

and learning a policy with the maximum entropy objective,

$$\max_\pi \mathbb{E}_{\mathbf{x}_t \sim \mathcal{D}} \mathbb{E}_{\pi(\mathbf{a}_t|\mathbf{x}_t)} \left[ Q(\mathbf{x}_t, \mathbf{a}_t) + \alpha \mathcal{H}(\pi(\cdot|\mathbf{x}_t)) \right]. \quad (3)$$

where states are sampled from a replay buffer $\mathcal{D}$ during training. In practice, SAC is often restricted to the use of policies where the entropy can be computed efficiently, *e.g.* a factorized Gaussian policy for continuous control environments. This allows random movements to occur as noise is added independently for each action dimension. Our proposed approach, on the other hand, introduces a structure in the exploration noise level.

## 2.2 World Models for Partially-Observed Environments

In many practically motivated settings, the agents only have access to certain observations, *e.g.* partial states, and the complete states must be inferred through observations. This can be modelled through the partially observed MDP (POMDP) graphical model, which encompasses a wide range of problem settings involving uncertainty. POMDP can be used to model uncertainty in the state, reward, or even the transition model itself (Åström, 1964). Here, the optimal policy must take into account these uncertainties, naturally becoming stochastic and may exhibit multi-modal behaviors (Todorov, 2006). Notationally, we only have access to observations $x_t \in \mathcal{X}$ with incomplete information, while the latent state $\mathbf{s}_t \in \mathcal{S}$ is unobserved, leading to a latent state transition distribution $p(\mathbf{s}_t|\mathbf{s}_{t-1}, \mathbf{a}_{t-1})$, observation distribution $p(\mathbf{x}_t|\mathbf{s}_t)$, and reward function $r_t(\mathbf{s}_t, \mathbf{a}_t)$.

In order to tackle this regime, people have resorted to learning world models (Deisenroth & Rasmussen, 2011; Ha & Schmidhuber, 2018) that attempt to learn a belief state conditioned on the history of observations and actions, typically viewed as performing variational inference on the POMDP, *i.e.* the world model is then responsible for tracking a belief state $\mathbf{s}_t$, which is update based on new observations through an inference model $q(\mathbf{s}_t|\mathbf{s}_{t-1}, \mathbf{a}_{t-1}, \mathbf{x}_t)$. The POMDP and the inference model are often jointly trained by maximizing a variational bound on the likelihood of observations,

$$\log p(\mathbf{x}_{1:T}|\mathbf{a}_{1:T}) \geq \mathbb{E}_q \left[ \sum_{t=1}^{T} \log p(\mathbf{x}_t|\mathbf{s}_t) - D_{\mathrm{KL}}(q(\mathbf{s}_t|\mathbf{s}_{t-1}, \mathbf{a}_{t-1}, \mathbf{x}_t) \| p(\mathbf{s}_t|\mathbf{s}_{t-1}, \mathbf{a}_{t-1})) \right]. \quad (4)$$

The world model is then typically paired with a policy that makes use of the belief state to take actions, *i.e.* $\pi(\mathbf{a}_t|\mathbf{s}_t)$ with $\mathbf{s}_t \sim q(\mathbf{s}_t|\mathbf{a}_{<t}, \mathbf{x}_{\leq t})$, as the assumption is that the posterior distribution over $\mathbf{s}_t$ contains all the information we have so far regarding the current state.

## 3 Stochastic Marginal Actor-Critic (SMAC)

We now discuss the use of latent variables for parameterizing policy distributions, and how these appear naturally under the use of a world model. We discuss the difficulties in handling latent variable policies in reinforcement learning, and derive cost-efficient low-variance stochastic estimators for marginalizing the latent state. Finally, we put it all together in an actor-critic framework.

### 3.1 Latent variable policies

We advocate the use of latent variables for constructing policy distributions as an effective yet simple way of increasing flexibility. This generally adds minimal changes to existing stochastic policy algorithms. Starting with the MDP setting, a latent variable policy (LVP) can be expressed as

$$\pi(\mathbf{a}_t|\mathbf{x}_t) := \int \pi(\mathbf{a}_t|\mathbf{s}_t) q(\mathbf{s}_t|\mathbf{x}_t) \, \mathrm{d}\mathbf{s}_t, \quad (5)$$

where $\mathbf{s}_t$ is a latent variable conditioned on the current observation. In the MDP setting, the introduction of a latent $q(\mathbf{s}_t|\mathbf{x}_t)$ mainly increases the expressiveness of the policy. This thus allows the policy to better capture a wider frontier of optimal actions, which can be especially helpful during the initial exploration when we lack information regarding future rewards. We discuss extensions to POMDPs shortly in the following section, where the policy is conditioned on a history of observations.

For parameterization, we use factorized Gaussian distributions for both $\pi(\mathbf{a}_t|\mathbf{s}_t)$ and $q(\mathbf{s}_t|\mathbf{x}_t)$. Firstly, this results in a latent variable policy that is computationally efficient: sampling and density evaluations both remain cheap. Furthermore, this allows us to build upon existing stochastic policy algorithms and architectures that have been used with a single Gaussian distribution, by simply adding a new stochastic node $\mathbf{s}_t$. Secondly, we can show that this is also a sufficient parameterization: with standard neural network architectures, a latent variable policy can universally approximate any distribution if given sufficient capacity. Intuitively, it is known that a mixture of factorized Gaussian is universal as the number of mixture components increases, and we can roughly view a latent variable model with Gaussian-distributed $\pi$ and $q$ as an infinite mixture of Gaussian distributions.

**Proposition 1.** *For any $d$-dimensional continuous distribution $p^*(x)$, there exist a sequence of two-level latent variable model $p_n(x) = \int p_n(x|z)p_n(z)\,\mathrm{d}z, n \in \mathbb{N}_+$ that converge to it, where both $p_n(x|z)$ and $p_n(z)$ are factorized Gaussian distributions with mean and variance parameterized by neural networks.*

Proof can be found in Appendix D.1.

### 3.1.1 WORLD MODELS INDUCE LATENT VARIABLE POLICIES

Perhaps unsurprisingly, latent variables already exist as part of many reinforcement learning works. In particular, in the construction of probabilistic world models, used when the environment is highly complex or only partially observable. Some works only use intermediate components of the world model as a deterministic input to their policy distribution (*e.g.* Lee et al. (2020a)), disregarding the distributional aspect, while other approaches use iterative methods for producing an action (*e.g.* Hafner et al. (2020)). We instead simply view the world model for what it is—a latent state inference model—which naturally induces a latent variable policy,

$$\pi(\mathbf{a}_t|\mathbf{a}_{<t}, \mathbf{x}_{\leq t}) = \int \pi(\mathbf{a}_t|\mathbf{s}_t)q(\mathbf{s}_t|\mathbf{a}_{<t}, \mathbf{x}_{\leq t})\,\mathrm{d}\mathbf{s}_t. \tag{6}$$

This follows the form of Equation 5 where the context includes the entire history, *i.e.* $\mathbf{h}_t = (\mathbf{a}_{<t}, \mathbf{x}_{\leq t})$. Note that $\pi(\mathbf{a}_t|\mathbf{s}_t)$ conditions only on the current latent state due to a Markov assumption typically used in existing world models (see Figure 2), though our algorithms easily extend to non-Markov settings as well. Furthermore, this marginalizes over the full latent history due to the recurrence,

$$q(\mathbf{s}_t|\mathbf{a}_{<t}, \mathbf{x}_{\leq t}) = \int q(\mathbf{s}_t|\mathbf{s}_{t-1}, \mathbf{a}_{t-1}, \mathbf{x}_t)q(\mathbf{s}_{t-1}|\mathbf{a}_{<t-1}, \mathbf{x}_{\leq t-1})\,\mathrm{d}\mathbf{s}_{t-1}, \tag{7}$$

which when recursively applied, we can see that the belief state $\mathbf{s}_t$—and hence the policy—marginalizes over the entire history of belief states. A more thorough discussion is in Appendix B.2.

Our approaches for handling latent variables are agnostic to what $q$ conditions on, so to unify and simplify notation, we use the shorthand $\mathbf{h}_t \triangleq (\mathbf{a}_{<t}, \mathbf{x}_{\leq t})$ to denote history information. This subsumes the MDP setting, where $q(\mathbf{s}_t|\bar{\mathbf{h}}_t)$ is equivalent to $q(\mathbf{s}_t|\mathbf{x}_t)$ due to Markovian conditional independence.

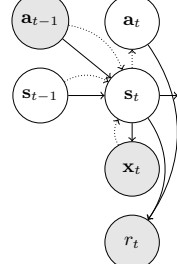

Figure 2: Graphical model of POMDP (*solid*), world model, and induced latent variable policy (*dashed*).

### 3.2 MAXENT RL IN THE PRESENCE OF LATENT VARIABLES

The presence of latent variables makes training with the maximum entropy objective (equations 1 and 3) difficult. Firstly, it requires an accurate estimation of the entropy term, and the entropy of a latent variable model is notoriously hard to estimate due to the intractability of marginalization (Paninski, 2003; Lim et al., 2020). Secondly, the use of latent variables results in an increase in gradient variance, which we remedy with variance reduction methods at a negligible cost. Finally, the appearance of

latent variables can also be used within the $Q$-function to better aggregate uncertainty. For each, we derive principled methods for handling latent variables, while the end result is actually fairly simple and only adds a minimal amount of extra cost compared to non-latent variable policies.

### 3.2.1 ESTIMATING THE MARGINAL ENTROPY

An immediate consequence of using latent variables is that the entropy, or marginal entropy, becomes intractable, due to the log-probability being intractable, *i.e.*

$$\mathcal{H}(\pi(\cdot|\mathbf{h}_t)) = \mathbb{E}_{\pi(\mathbf{a}_t|\mathbf{h}_t)}\left[-\log \int \pi(\mathbf{a}_t|\mathbf{s}_t)q(\mathbf{s}_t|\mathbf{h}_t)\,\mathrm{d}\mathbf{s}_t\right]. \tag{8}$$

**Failure cases of naïve entropy estimation** Applying methods developed for amortized variational inference (Kingma & Welling, 2013; Burda et al., 2016) can result in a bound on the entropy that is in the wrong direction. For instance, the standard evidence lower bound (ELBO) results in an entropy estimator,

$$\widetilde{\mathcal{H}}_{\text{naïve}}(\mathbf{h}_t) \triangleq \mathbb{E}_{\pi(\mathbf{a}_t|\mathbf{h}_t)}\mathbb{E}_{r(\mathbf{s}_t|\mathbf{a}_t,\mathbf{h}_t)}\left[-\log \pi(\mathbf{a}_t|\mathbf{s}_t) + \log \tilde{q}(\mathbf{s}_t|\mathbf{a}_t,\mathbf{h}_t) - \log q(\mathbf{s}_t|\mathbf{h}_t)\right], \tag{9}$$

where $\tilde{q}$ is any variational distribution, for example setting $\tilde{q}(\mathbf{s}_t|\mathbf{a}_t,\mathbf{h}_t) = q(\mathbf{s}_t|\mathbf{h}_t)$. Adopting this naïve estimator will result in maximizing an *upper* bound on the MaxEnt RL objective, which we can see by writing out the error,

$$\widetilde{\mathcal{H}}_{\text{naïve}}(\mathbf{h}_t) = \mathcal{H}(\pi(\cdot|\mathbf{h}_t)) + \mathbb{E}_{\pi(\mathbf{a}_t|\mathbf{h}_t)}\left[D_{\text{KL}}(\tilde{q}(\mathbf{s}_t|\mathbf{a}_t,\mathbf{h}_t)\|q(\mathbf{s}_t|\mathbf{a}_t,\mathbf{h}_t))\right]. \tag{10}$$

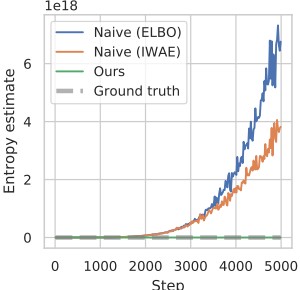

Therefore, replacing the entropy in the MaxEnt RL objective (Equation 1) with $\widetilde{\mathcal{H}}_{\text{naïve}}$ will lead to maximizing the error—*i.e.* the KL divergence—incentivizing the variational distribution to be as *far* as it can from the true posterior $q(\mathbf{s}_t|\mathbf{a}_t,\mathbf{h}_t)$. Furthermore, this error is unbounded so it may become arbitrarily large without actually affecting the true entropy we want to be maximizing, $\mathcal{H}(\pi(\cdot|\mathbf{h}_t))$, which leads to serious numerical instability issues. In Figure 3, we show the results from a preliminary experiment where this approach to entropy estimation during policy optimization led to extremely large values (scale of $10^{18}$), significantly overestimating the true entropy, and resulted in policies that did not learn. More details are in Appendix C. To overcome this overestimation issue, we propose the following method for achieving accurate estimation.

Figure 3: Training with naïve entropy estimators results in extremely loose upper bounds.

**Lower bounding the marginal entropy with nested estimator** To be amenable to entropy maximization, we must construct a lower bound estimator of the marginal entropy. For this, inspired by advances in hierarchical inference (Yin & Zhou, 2018; Sobolev & Vetrov, 2019), a method to estimate the marginal entropy (Equation 8) via a *lower* bound can be obtained. Specifically, for any $K \in \mathbb{N}$, we define

$$\widetilde{\mathcal{H}}_K(\mathbf{h}_t) \triangleq \mathbb{E}_{\mathbf{a}_t\sim\pi(\mathbf{a}_t|\mathbf{h}_t)}\mathbb{E}_{\mathbf{s}_t^{(0)}\sim p(\mathbf{s}_t|\mathbf{a}_t,\mathbf{h}_t)}\mathbb{E}_{\mathbf{s}_t^{(1:K)}\sim q(\mathbf{s}_t|\mathbf{h}_t)}\left[-\log\left(\frac{1}{K+1}\sum_{k=0}^{K}\pi\left(\mathbf{a}_t|\mathbf{s}_t^{(k)}\right)\right)\right]. \tag{11}$$

where $p(\mathbf{s}_t|\mathbf{a}_t,\mathbf{h}_t)$ is the (unknown) posterior of the policy distribution; however, we can easily sample from this by first sampling $\mathbf{s}_t^{(0)}$ then sample $\mathbf{a}_t$ conditioned on $\mathbf{s}_t^{(0)}$. This results in a nested estimator where we effectively sample $K + 1$ times from $q(\mathbf{s}_t|\mathbf{h}_t)$, use only the first latent variable $s_t^{(0)}$ for sampling the action, while using all the latent variables to estimate the marginal entropy. Note that this is *not* equivalent to replacing the expectation inside the logarithm with independent samples, which would correspond to an IWAE estimator (Burda et al., 2016). Equation 11 results in a nested estimator that is monotonically increasing in $K$, which in the limit, becomes an unbiased estimator of the marginal entropy, *i.e.* $\widetilde{\mathcal{H}}_K(\mathbf{h}_t) \leq \mathcal{H}[\pi(\cdot|\mathbf{h}_t)]$, $\widetilde{\mathcal{H}}_K(\mathbf{h}_t) \leq \widetilde{\mathcal{H}}_{K+1}(\mathbf{h}_t)$ and $\lim_{K\to\infty}\widetilde{\mathcal{H}}_K(\mathbf{h}_t) = \mathcal{H}[\pi(\cdot|\mathbf{h}_t)]$. Thus, replacing the marginal entropy with $\mathcal{H}_K$ results in maximizing a tight lower bound on the MaxEnt RL objective, and is much more numerically stable in practice. Proofs for these results are in Appendix D. In practice, we find that using reasonable values for $K$ does not increase computation times since sampling multiple times is easily done in parallel, and the evaluation of $\pi(\mathbf{a}_t|\mathbf{s}_t)$ is cheap relative to other components such as the world model.

### 3.2.2 Variance reduction with antithetic multi-level Monte Carlo

While latent variable policies can optimize for the MaxEnt RL objective better in expectation, its reliance on stochastic estimation techniques introduces additional gradient variance. This higher variance actually results in poorer sample efficiency, negating any gains obtained from using a more flexible distribution. In particular, it has been shown that multi-sample estimators like Equation 11 can result in more noise than signal as $K$ increases (Rainforth et al., 2018a). To remedy this, we adopt a simple yet reliable variance reduction method referred to as antithetic multi-level Monte Carlo (MLMC). While this method has been used in simulations of stochastic differential equations (Giles, 2008; Giles & Szpruch, 2014) and more recently, in variational inference (Ishikawa & Goda, 2021; Shi & Cornish, 2021), it has not yet seen uses in the context of reinforcement learning.

Applying MLMC to the estimator in Equation 11, we have

$$\widetilde{\mathcal{H}}_K^{\text{MLMC}} = \sum_{\ell=0}^{\lfloor \log_2(K) \rfloor} \Delta \widetilde{\mathcal{H}}_{2^\ell}, \quad \text{where} \ \ \Delta \widetilde{\mathcal{H}}_{2^\ell} = \begin{cases} \widetilde{\mathcal{H}}_1 & \text{if } \ell = 0, \\ \widetilde{\mathcal{H}}_{2^\ell} - \frac{1}{2} \left( \widetilde{\mathcal{H}}_{2^{\ell-1}}^{(a)} + \widetilde{\mathcal{H}}_{2^{\ell-1}}^{(b)} \right) & \text{otherwise.} \end{cases} \tag{12}$$

At the $\ell$-th level, after we have generated $2^\ell$ i.i.d. samples, we use half to compute $\widetilde{\mathcal{H}}_{2^{\ell-1}}^{(a)}$, the other half to compute $\widetilde{\mathcal{H}}_{2^{\ell-1}}^{(b)}$, and all of the samples to compute $\widetilde{\mathcal{H}}_{2^\ell}$. This antithetic sampling scheme is a key ingredient in reducing variance, and can achieve the optimal computational complexity for a given accuracy (Ishikawa & Goda, 2021). We compute all the $\Delta \widetilde{\mathcal{H}}_{2^\ell}$ terms in parallel in our implementation, so there is an almost negligible additional cost compared to $\widetilde{\mathcal{H}}_K$. The only consideration involved for using $\widetilde{\mathcal{H}}_K^{\text{MLMC}}$ is that $K$ should be a power of two.

### 3.2.3 Estimating the marginal Q-function

Under the POMDP setting, we aim to build a Q-function upon the inferred belief states $\mathbf{s}_t$ as these contain the relevant dynamics and reward information. However, while most existing approaches such as Lee et al. (2020a); Hafner et al. (2020) only take one sample of the inferred latents as input of the Q-function, we propose marginalizing out the latent distribution in the critic calculation. This can be seen by interpreting the Q-function through the probabilistic inference framework in Levine (2018), where the reward function is viewed as the log-likelihood of observing a binary optimality random variable $\mathcal{O}$, i.e. satisfying $p(\mathcal{O}_t = 1|\mathbf{s}_t, \mathbf{a}_t) \propto \exp(r(\mathbf{s}_t, \mathbf{a}_t))$. As a result, the Q-function is equivalent to $Q(\mathbf{s}_t, \mathbf{a}_t) = \log p(\mathcal{O}_{t:T}|\mathbf{s}_t, \mathbf{a}_t)$. Since our latent belief state represents the uncertainty regarding the system in the context of POMDPs, including the current state and unseen rewards, we propose marginalizing the value function over the belief state. Hence, through this probabilistic interpretation, the marginal Q-function is related to the Q-function over latent states through

$$Q(\mathbf{h}_t, \mathbf{a}_t) = \log \int p(\mathcal{O}_{t:T}|\mathbf{s}_t, \mathbf{a}_t) q(\mathbf{s}_t|\mathbf{h}_t) \, \mathrm{d}\mathbf{s}_t = \log \int \exp \{Q(\mathbf{s}_t, \mathbf{a}_t)\} q(\mathbf{s}_t|\mathbf{h}_t) \, \mathrm{d}\mathbf{s}_t. \tag{13}$$

Given this, we hence propose the following estimator to be used during policy optimization,

$$Q(\mathbf{h}_t, \mathbf{a}_t) \approx \widetilde{Q}_K(\mathbf{h}_t, \mathbf{a}_t) \triangleq \log \left( \frac{1}{K} \sum_{k=1}^{K} \exp \left\{ Q(\mathbf{s}_t^{(k)}, \mathbf{a}_t) \right\} \right), \quad \mathbf{s}_t^{(1:K)} \sim q(\mathbf{s}_t|\mathbf{h}_t), \tag{14}$$

where $Q(\mathbf{s}_t, \mathbf{a}_t)$ is trained to satisfy the soft Bellman equation in Equation 2. A closely related approach is from Lee et al. (2020a) who similarly trains a Q-function on latent states; however, they directly use $Q(\mathbf{s}_t, \mathbf{a}_t)$ during policy optimization, which is a special case with $K = 1$, whereas using $K > 1$ results in a closer approximation to the marginal Q-function. We found this construction for marginalizing the Q-function to be useful mainly when used in conjunction with a world model.

### 3.3 Stochastic Marginal Actor-Critic (SMAC)

While the above methods can each be applied to general MaxEnt RL algorithms, we instantiate a concrete algorithm termed Stochastic Marginal Actor-Critic (SMAC). SMAC is characterized by the use of a latent variable policy and maximizes a lower bound to a marginal MaxEnt RL objective. Specifically, we use the same method as SAC to train $Q(\mathbf{s}_t, \mathbf{a}_t)$ on latent states, but we train the policy using a low-variance debiased objective for taking into account latent state marginalization,

$$\max_\pi \mathbb{E}_{\mathbf{h}_t \sim \mathcal{D}, \mathbf{a}_t \sim \pi} \left[ \widetilde{Q}_K(\mathbf{h}_t, \mathbf{a}_t) + \alpha \widetilde{\mathcal{H}}_K^{\text{MLMC}}(\mathbf{h}_t) \right]. \tag{15}$$

We train the inference model $q(\mathbf{s}_t|\mathbf{h}_t)$ with standard amortized variational inference (Equation 4), and we train only $\pi(\mathbf{a}_t|\mathbf{s}_t)$ using the objective in Equation 15. When not used with a world model, we train both $\pi(\mathbf{a}_t|\mathbf{s}_t)$ and $q(\mathbf{s}_t|\mathbf{h}_t)$ using Equation 15. See Algorithms 1 and 2 for a summary of the training procedures, and more details regarding implementation for SMAC in Appendix B.

## 4  RELATED WORK

**Maximum entropy reinforcement learning**   Prior works have demonstrated multiple benefits of MaxEnt RL, including improved exploration (Han & Sung, 2021), regularized behaviors (Neu et al., 2017; Vieillard et al., 2020a), better optimization property (Ahmed et al., 2018), and stronger robustness (Eysenbach & Levine, 2022). Generally, policies optimizing the MaxEnt RL objective sample actions that are proportional to the exponentiated reward, and alternatively can be viewed as a noise injection procedure for better exploration (Attias, 2003; Ziebart, 2010; Haarnoja et al., 2017; Nachum et al., 2017; Levine, 2018; Abdolmaleki et al., 2018; Haarnoja et al., 2018b; Vieillard et al., 2020b; Pan et al., 2022; 2023; Lahlou et al., 2023). However, this noise injection is commonly done directly in action space, leading to only local perturbations, whereas we inject noise through a nonlinear mapping.

**Latent variable modeling**   The usage of latent variable models originates from graphical models (Dayan et al., 1995; Hinton et al., 2006) and has been recently popularized in generative modeling (Kingma & Welling, 2013; Rezende et al., 2014; Zhang et al., 2021a; 2022b;a). The estimation of the log marginal probability and marginal entropy has long been a central problem in Bayesian statistics and variational inference (Newton, 1994; Murray & Salakhutdinov, 2008; Nowozin, 2018; Ishikawa & Goda, 2021; Malkin et al., 2022). However, most of these works consider a lower bound on the log marginal probability for variational inference, which is not directly applicable to maximum entropy as discussed in Section 3.2.1. A few works have proposed upper bounds (Sobolev & Vetrov, 2019; Dieng et al., 2017) or even unbiased estimators (Luo et al., 2020), and while we initially experimented with a couple of these estimators, we found that many results in high gradient variance and ultimately identified an approach based on hierarchical inference technique for its efficiency and suitability in RL.

**Latent structures in POMDPs and world models**   Settings with only partial observations are natural applications for probabilistic inference methods, which help learn latent belief states from observational data. As such, variational inference has been adopted for learning sequential latent variable models (Ghahramani & Hinton, 2000; Krishnan et al., 2015; Fraccaro et al., 2016; Karl et al., 2017; Singh et al., 2021). One paradigm is to use the learned recurrent model to help model-free RL algorithms (Wahlstrom et al., 2015; Tschiatschek et al., 2018; Buesing et al., 2018; Igl et al., 2018; Gregor et al., 2019; Han et al., 2020). Another approach is to use world models for solving POMDP and building model-based RL agents  (Deisenroth & Rasmussen, 2011; Hausknecht & Stone, 2015; Watter et al., 2015; Zhang et al., 2019; Hafner et al., 2019; 2020; 2021; Nguyen et al., 2021; Chen et al., 2022) due to their planning capabilities. It is also sometimes the case that the world model is treated mainly as a representation, without much regard for the measure of uncertainty (Ha & Schmidhuber, 2018; Schrittwieser et al., 2020; Amos et al., 2021; Hansen et al., 2022).

## 5  EXPERIMENTS

We evaluate SMAC on a series of diverse continuous control tasks from DeepMind Control Suite (DMC; Tassa et al. (2018)). These tasks include challenging cases in the sense of having sparse rewards, high dimensional action space, or pixel observations. We also perform a preliminary unit test with a multi-modal reward in Appendix B.1.

### 5.1  STATE-BASED CONTINUOUS CONTROL ENVIRONMENTS

**Setting**   We first compare SMAC with SAC and TD3 (Fujimoto et al., 2018) baselines on a variety of state-based environments to demonstrate the advantage of latent variable policies. We show eight environments in Figure 4, and leave more results in Appendix due to space limitations.

**Results**   We find that even in the simple MDP setting, we can improve upon SAC by simply introducing a latent variable. Specifically, our method is almost never worse than SAC, implying that the extra gradient variance from the entropy estimation is not incurring a penalty in terms of sample efficiency.

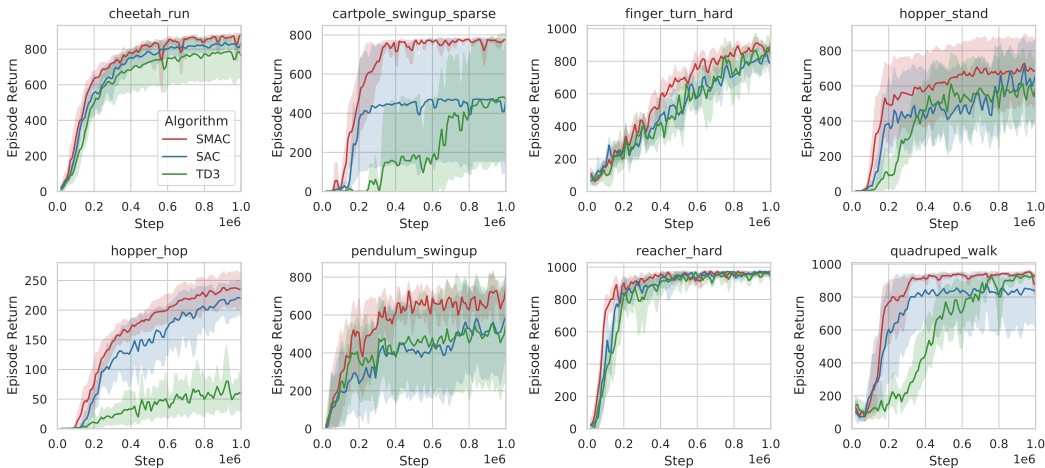

Figure 4: Experiments on eight DMC environments where agents are given state-based inputs. The SMAC approach improves upon SAC with better exploration and more robust training.

By being able to track a wider frontier of optimal action trajectories, SMAC can be more robust to find optimal policies, particularly when the reward is sparse (*e.g.*, `cartpole_swingup_sparse`).

**Comparison with other probabilistic modeling approaches**    We further conduct extensive empirical comparisons with other probabilistic policy modeling methods including normalizing flow and mixture-of-experts (Ren et al., 2021) based SAC methods in Figure 11. Our proposed SMAC generally achieves the best sample efficiency on almost all environments. Due to limited space, we defer related discussion to Appendix C.

**Marginalization**    Marginalizing over the latent state has a significant effect on training, though this often exhibits a diminishing rate, suggesting that using reasonable number of particles is sufficient. Figure 5 shows this behavior for the `quadruped_walk` task.

**Variance-reduced updates**    We find that the use of MLMC as a variance reduction tool is crucial for the latent variable policy to perform well in some difficult environments such as the `quadruped_escape` task. Figure 5b shows that using MLMC clearly reduces variance and makes training much more robust, whereas using only the nested estimator performs closer to the baseline SAC (see comparison in Figure 10).

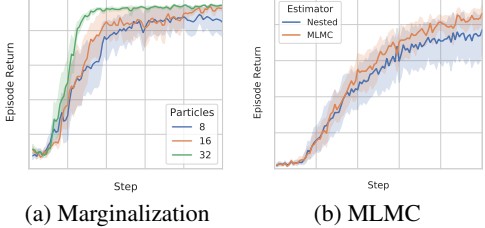

(a) Marginalization          (b) MLMC

Figure 5: Ablation experiments.

## 5.2   Pixel-based continuous control environments

**Setting**    We next compare different algorithms on a series of DMC environments with pixel-based input. Since this task is much more challenging, and pixels only provide partial observability, we make use of a world model (as described in Section 2.2) to supplement our algorithm. We use the recurrent state-space model (RSSM) architecture from Hafner et al. (2019) as the world model. We refer to this baseline as "Latent-SAC" and follow the practice in Wang et al. (2022), which samples from the belief distribution $q(\mathbf{s}_t|\mathbf{a}_{<t}, \mathbf{x}_{\leq t})$ and directly trains using SAC on top of the belief state. A closely related work, SLAC (Lee et al., 2020a), only uses $\mathbf{s}_t$ as input to a learned Q-function, while the policy does not use $\mathbf{s}_t$ and instead uses intermediate layers of the world model as input. Finally, we also compare to Dreamer, a model-based RL (MBRL) algorithm that performs rollouts on the dynamics model (Hafner et al., 2020). This iterative procedure results in a higher computational cost as it requires iteratively sampling from the belief state and differentiating through the rollouts. In contrast, our proposed SMAC aggregates samples from the current belief state and does not require

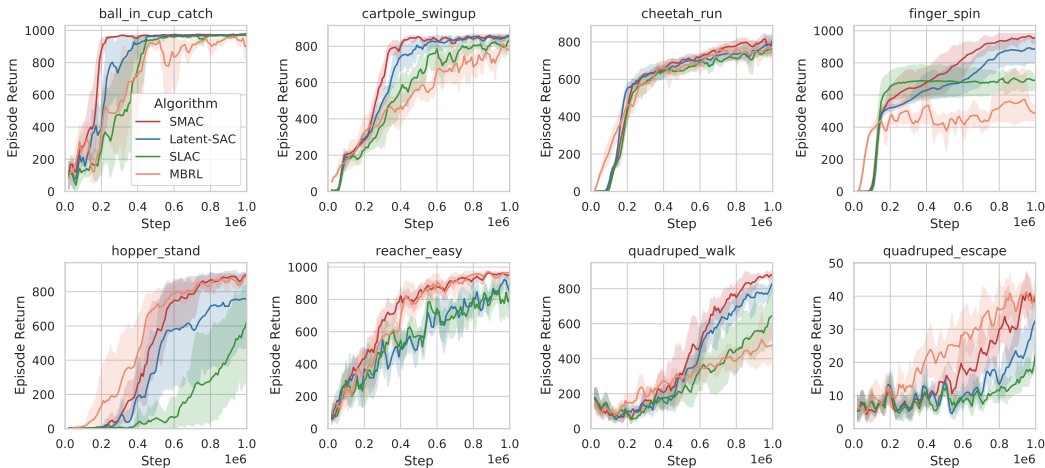

Figure 6: Experiments on eight DMC environments where agents are given pixel-based inputs. The proposed SMAC approach achieves better expected return than similar Latent-SAC and SLAC baselines. Notice that our method does *not* involve any planning, but still achieves comparable (sometimes even better) performance to the model-based RL algorithm.

Table 1: Experiments with noisy observations. We experiment with two different perturbation level for two kinds of noise. GAUSSIAN PERTURBATION adds independent white noise to the pixel images while SENSOR MISSING randomly turns a portion of the pixels to black. SMAC is trained with a world model while L-SAC denotes a SAC baseline trained with a world model.

| | GAUSSIAN PERTURBATION | | | | SENSOR MISSING | | | |
| NOISE | SMALL | | LARGE | | SMALL | | LARGE | |
| METHOD | L-SAC | SMAC | L-SAC | SMAC | L-SAC | SMAC | L-SAC | SMAC |
|---|---|---|---|---|---|---|---|---|
| FINGER | $912 \pm 138$ | $\mathbf{959} \pm 30$ | $880 \pm 109$ | $\mathbf{924} \pm 43$ | $933 \pm 63$ | $\mathbf{955} \pm 36$ | $921 \pm 44$ | $921 \pm 52$ |
| HOPPER | $571 \pm 411$ | $\mathbf{731} \pm 407$ | $516 \pm 385$ | $\mathbf{702} \pm 391$ | $721 \pm 16$ | $\mathbf{872} \pm 62$ | $703 \pm 7$ | $\mathbf{866} \pm 61$ |
| REACHER | $869 \pm 28$ | $\mathbf{928} \pm 10$ | $782 \pm 89$ | $\mathbf{925} \pm 92$ | $883 \pm 112$ | $\mathbf{937} \pm 89$ | $854 \pm 169$ | $\mathbf{925} \pm 59$ |

differentiating through the dynamics model. For training, we follow Hafner et al. (2020) and repeat each action 2 times. We show the comparison on eight tasks in Figure 6, and again relegate more results to the Appendix due to space constraints.

**Results**    Comparing SMAC to the Latent-SAC baseline, we again find that we can often find an optimal policy with fewer environment interactions. We find that SLAC and Latent-SAC are roughly on par, while SLAC can also some times perform worse, as their policy does not condition on the latent state. The model-based approach has widely-variable performance when compared to the actor-critic approaches. Interestingly, in most of the environments where the model-based approach is performing well, we find that SMAC can often achieve comparable performance, even though it does not make use of planning. Overall, we find that our method improves upon actor-critic approaches and bridges the gap to planning-based approaches.

**Robustness to noisy observations**    While the pixel-based setting already provides partial observations, we test the robustness of our approach in settings with higher noise levels. In Table 1 we report the episodic rewards of both SMAC and a SAC baseline on three environments (finger_spin, hopper_stand, and reacher_easy) under noisy perturbations and missing pixels (Meng et al., 2021). We find that SMAC behaves more robustly than the baseline across almost all settings.

**Efficiency**    Despite the extra estimation procedures, SMAC does not incur significant computational costs as we can compute all terms in the estimators in parallel. Tested with an NVIDIA Quadro GV100 on the pixel-based environments, our SMAC implementation does 60 frames per second (FPS)

on average, almost the same training speed compared to Latent-SAC (63 FPS), whereas differentiating through a single rollout over the dynamics model already reduces to 51 FPS (roughly 20% slower).

## 6 CONCLUSION

We propose methods for better handling of latent variable policies under the MaxEnt RL framework, centered around cost-efficient computation and low-variance estimation, resulting in a tractable algorithm SMAC when instantiated in the actor-critic framework. We find that SMAC can better make use of the belief state than competing actor-critic methods and can more robustly find optimal policies, while adding only very minimal amounts of extra compute time.

## ACKNOWLEDGEMENT

The authors would like to thank Zhixuan Lin, Tianwei Ni, Chinwei Huang, Brandon Amos, Ling Pan, Max Schwarzer, Yuandong Tian, Tianjun Zhang, Shixiang Gu, and anonymous reviewers for helpful discussions. Dinghuai also expresses gratitude towards his fellow interns at FAIR for creating a lot of joyful memories during the summer in New York City.

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

# A NOTATIONS

| Symbol | Description |
|---|---|
| $\mathbf{x}_t$ | Environment state (human designed feature or pixel image) at time $t$ |
| $\mathbf{a}_t$ | Action at time $t$ |
| $\mathbf{h}_t$ | History up to time $t$, defined as $(\mathbf{a}_{<t}, \mathbf{x}_{\leq t})$ |
| $r$ | Reward |
| $\pi$ | Policy (distribution over action) |
| $Q$ | Q-function |
| $\widetilde{Q}$ | Q-function estimator |
| $\gamma$ | Discount factor |
| $\mathcal{O}$ | Optimality binary random variable as defined in Levine (2018) |
| $\mathcal{S}$ | Domain of state |
| $\mathcal{A}$ | Domain of action |
| $\mathcal{H}(\cdot)$ | Entropy |
| $\widetilde{\mathcal{H}}(\cdot)$ | Entropy estimator |
| $\alpha$ | Temperature for MaxEnt RL |
| $\mathbf{s}_t$ | Latent variable / inferred belief state |
| $q(\mathbf{s}_t\|\mathbf{x}_t)$ | Belief distribution over inferred states from observation $\mathbf{x}_t$ |
| $q(\mathbf{s}_t\|\mathbf{a}_{<t}, \mathbf{x}_{\leq t})$ | Belief distribution over inferred states from past data |
| $q(\mathbf{s}_t\|\mathbf{h}_t)$ | Unifies notation for the above two |
| $\pi(\mathbf{a}_t\|\mathbf{s}_t)$ | Policy conditioned on latent variable $\mathbf{s}_t$ |
| $\pi(\mathbf{a}_t\|\mathbf{x}_t)$ | Latent variable policy, equals $\int \pi(\mathbf{a}_t\|\mathbf{s}_t)q(\mathbf{s}\|\mathbf{x})\,\mathrm{d}\mathbf{s}$ |
| $\pi(\mathbf{a}_t\|\mathbf{a}_{<t}, \mathbf{x}_{\leq t})$ | Latent variable policy, equals $\int \pi(\mathbf{a}_t\|\mathbf{s}_t)q(\mathbf{s}_t\|\mathbf{a}_{<t}, \mathbf{x}_{\leq t})\,\mathrm{d}\mathbf{s}_t$ |
| $\pi(\mathbf{a}_t\|\mathbf{h}_t)$ | Unifies notation for the above two |
| $p(\mathbf{x}_t\|\mathbf{s}_t)$ | Observation model in the world model |
| $p(r_t\|\mathbf{s}_t)$ | Reward model in the world model |
| $p(\mathbf{s}_{t+1}\|\mathbf{s}_t, \mathbf{a}_t)$ | Transition model, also the prior of a world model |
| $q(\mathbf{s}_t\|\mathbf{s}_{t-1}, \mathbf{a}_{t-1}, \mathbf{x}_t)$ | Inferred posterior dynamics model of learned world model |

# B ADDITIONAL DETAILS REGARDING METHODOLOGY

## B.1 MULTI-MODALITY OF LATENT VARIABLE POLICIES

While a latent variable policy can theoretically model any distribution (see Proposition 1), training this policy can still be difficult, especially if the true reward is actually multi-modal. Here, we test in a controlled setting, whether our method can truly recover a multi-modal policy.

A standard interpretation of the MaxEnt RL objective is as a reverse KL objective (Levine, 2018),

$$\max_\pi \mathbb{E}_{p(\mathbf{x})\pi(\mathbf{a}|\mathbf{x}))} \left[ r(\mathbf{x}, \mathbf{a}) - \alpha \log \pi(\mathbf{a}|\mathbf{x}) \right] \tag{16}$$

$$\Leftrightarrow \max_\pi \mathbb{E}_{p(\mathbf{x})\pi(\mathbf{a}|\mathbf{x}))} \left[ \frac{r(\mathbf{x}, \mathbf{a})}{\alpha} - \log \pi(\mathbf{a}|\mathbf{x}) \right] \tag{17}$$

$$\Leftrightarrow \max_\pi \mathbb{E}_{p(\mathbf{x})\pi(\mathbf{a}|\mathbf{x}))} \left[ \log \exp \left\{ \frac{r(\mathbf{x}, \mathbf{a})}{\alpha} \right\} - \log \pi(\mathbf{a}|\mathbf{x}) \right] \tag{18}$$

$$\Leftrightarrow \min_\pi \mathbb{E}_{p(\mathbf{x})} \left[ D_{\mathrm{KL}} \left( \pi(\mathbf{a}|\mathbf{x}) \| p^*(\mathbf{a}|\mathbf{x}) \right) \right] \tag{19}$$

where $p^*(\mathbf{a}|\mathbf{x}) \propto \exp \left\{ \frac{r(\mathbf{x},\mathbf{a})}{\alpha} \right\}$, *i.e.*, a target distribution defined by the exponentiated reward function and annealed with temperature $\alpha$.

Despite the ubiquity of the reverse KL objective—such as appearing in standard posterior inference—the training of latent variable models for this objective is still relatively under-explored due to the difficulty in estimating this objective properly. Luo et al. (2020) showed that using improper bounds on the objective can lead to catastrophic failure, but only showed successful training for a unimodal target distribution, while Sobolev & Vetrov (2019) discussed proper bounds but did not perform such an experiment.

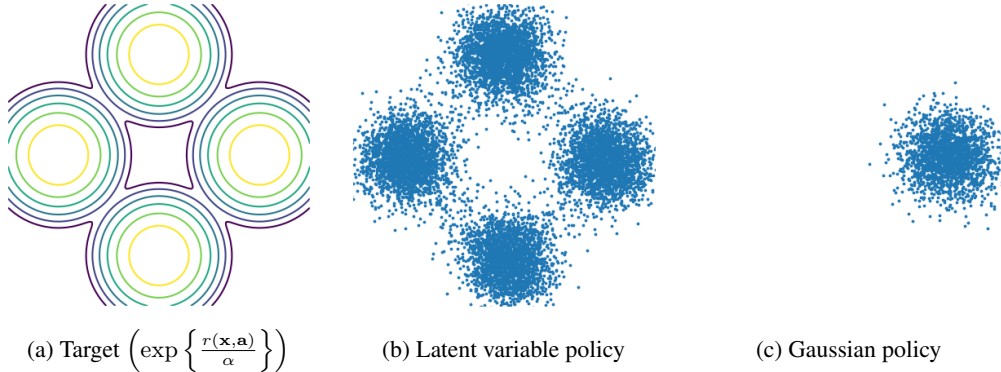

(a) Target $\left(\exp\left\{\frac{r(\mathbf{x},\mathbf{a})}{\alpha}\right\}\right)$   (b) Latent variable policy   (c) Gaussian policy

Figure 7: Optimizing a latent variable policy for a one-step multi-modal MaxEnt RL objective.

We experiment by setting a reward function that has multiple optimal actions. Using a sufficiently large $\alpha$ creates a target distribution with four modes (Figure 7a). In Figure 7b, we show that we can successfully learn a multi-modal distribution with a latent variable policy using the methods discussed in Section 3.2.1 and 3.2.2. On the other hand, a Gaussian policy can only capture one out of four modes (Figure 7c), with the exact mode depending on the random initialization.

## B.2 WORLD MODEL LEARNING

In Figure 8 we visualize the graphical model for the RSSM similarly with Hafner et al. (2019) described in Section 2.2. We use solid arrows to denote the generative machinery ($p$ in the following equations) and dotted arrows to denote the inference machinery ($q$ in the following equations). A variational bound for the likelihood on observed trajectory could be written as follows,

$$\log p(\mathbf{x}_{\leq T}, r_{\leq T}|\mathbf{a}_{\leq T}) \geq \mathbb{E}_{\mathbf{s}_{\leq T} \sim q}\left[\log p(\mathbf{x}_{\leq T}, r_{\leq T}, \mathbf{s}_{\leq T}|\mathbf{a}_{\leq T}) - \log q(\mathbf{s}_{\leq T}|\mathbf{x}_{\leq T}, \mathbf{a}_{\leq T})\right] \quad (20)$$

$$=\mathbb{E}_q\left[\sum_{t=1}^{T} \log p(\mathbf{x}_t|\mathbf{s}_t) + \log p(r_t|\mathbf{s}_t) + \log p(\mathbf{s}_t|\mathbf{s}_{t-1}, \mathbf{a}_{t-1}) - \log q(\mathbf{s}_t|\mathbf{s}_{t-1}, \mathbf{a}_{t-1}, \mathbf{x}_t)\right] \quad (21)$$

$$=\mathbb{E}_q\left[\sum_{t=1}^{T} \log p(\mathbf{x}_t|\mathbf{s}_t) + \log p(r_t|\mathbf{s}_t) - D_{\mathrm{KL}}\left(q(\mathbf{s}_t|\mathbf{s}_{t-1}, \mathbf{a}_{t-1}, \mathbf{x}_t)\|p(\mathbf{s}_t|\mathbf{s}_{t-1}, \mathbf{a}_{t-1})\right)\right]. \quad (22)$$

The world model / RSSM is then learned by maximizing Equation 22 with regard to parameters of $p(\mathbf{x}|\mathbf{s}), p(r|\mathbf{s}), q(\mathbf{s}_t|\mathbf{s}_{t-1}, \mathbf{a}_{t-1}, \mathbf{x}_t)$ and $p(\mathbf{s}_t|\mathbf{s}_{t-1}, \mathbf{a}_{t-1})$. Note that in Section 2.2 we omit the reward modeling part for simplicity. Due to the Markovian assumption on latent dynamics and the shorthand of $\mathbf{h}_t \triangleq (\mathbf{a}_{<t}, \mathbf{x}_{\leq t})$, we could also use $q(\mathbf{s}_t|\mathbf{h}_t)$ to denote $q(\mathbf{s}_t|\mathbf{s}_{t-1}, \mathbf{a}_{t-1}, \mathbf{x}_t)$.

## B.3 SMAC ALGORITHM

In this section, we present the algorithmic details of SMAC with and without world model in Algorithm 1 and Algorithm 2 respectively. Both of the two algorithms follow the commonly adopted off-policy actor-critic style and utilize a replay buffer to save data for the update of both the actor and critic networks (the dependency of buffer $\mathcal{D}$ is omitted in the algorithms). Our SMAC is based on SAC algorithm, whose critic is trained by minimizing TD error,

$$\mathcal{J}_Q = \left(Q(\mathbf{x}, \mathbf{a}) - \left(r + \gamma\bar{Q}(\mathbf{x}', \mathbf{a}') + \alpha\widetilde{\mathcal{H}}(\pi(\cdot|\mathbf{x}'))\right)\right)^2, \quad (23)$$

where $(\mathbf{x}, \mathbf{a}, r, \mathbf{x}') \sim \mathcal{D}$, $\mathbf{a}' \sim \pi(\cdot|\mathbf{x}')$, $\bar{Q}$ denotes a stop gradient operator and $\widetilde{\mathcal{H}}$ is an estimate for the policy entropy. In our case, we estimate the entropy of latent variable

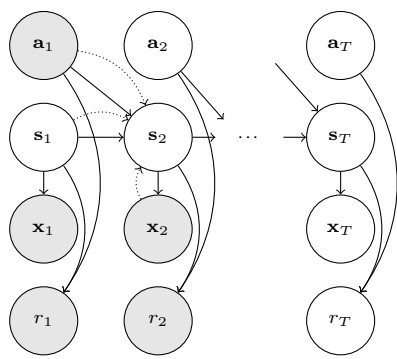

Figure 8: Graphical model of a POMDP (*solid*) and a world model (*dashed*).

policy with Equation 12 as discussed in Section 3.2. What's more, the actor is updated via minimizing

$$\mathcal{J}_\pi = -Q(\mathbf{x}, \mathbf{a}) - \alpha\widetilde{\mathcal{H}}(\pi(\cdot|\mathbf{x})), \tag{24}$$

where $\mathbf{x} \sim \mathcal{D}$ and $\mathbf{a} \sim \pi(\cdot|\mathbf{x})$, which is equivalent to $\mathbf{a} \sim \pi(\cdot|\mathbf{s})$, $\mathbf{s} \sim q(\mathbf{s}|\mathbf{x})$. In the algorithm box we omit the moving average of the critic network for simplicity, which is adopted as common-sense. We remark that SMAC has not much difference with SAC in the sense of RL algorithmic details, but mainly achieve improvement with the structured exploration behavior achieved from latent variable modeling. For SMAC in conjunction with a world model, we learn the critic network by minimizing TD error on the latent level,

$$\mathcal{J}_Q = \left(Q(\mathbf{s}, \mathbf{a}) - \left(r + \gamma\bar{Q}(\mathbf{s}', \mathbf{a}') + \alpha\widetilde{\mathcal{H}}(\pi(\cdot|\mathbf{s}'))\right)\right)^2, \tag{25}$$

whose terms can be seen as one sample estimate to each term in Equation 23. We also try to directly train the critic network on the observation level, but the empirical difference is negligible. As a result, we keep the latent level TD learning for simplicity's sake.

We next state a method to encourage exploration through conditional entropy minimization. Entropy in a latent variable policy (Equation 5) can be increased by either dispersing the probability density to other modes, or by increasing the entropy at a single mode. The latter corresponds to increasing the entropy of the conditional distribution $\pi(\mathbf{a}_t|\mathbf{s}_t)$, which can end up as a shortcut to increasing entropy and can result in spurious local minima during training. This is in fact a well-known issue that sampling-based objectives run into (Rainforth et al., 2018b; Midgley et al., 2022), resulting in a policy that explores the space of action trajectories at a slower pace. On the other hand, notice that in the proof of Proposition 1 in Section D.1 we require the decoder variance to be sufficiently small to be expressive, thus we propose remedying this issue for MaxEnt RL by adding a conditional entropy term to the objective:

$$\max_\pi \mathbb{E}_{p(\tau)}\left[\sum_{t=0}^\infty \gamma^t \left(r_t(\mathbf{x}_t, \mathbf{a}_t) + \alpha\mathcal{H}(\pi(\cdot|\mathbf{x}_t)) - \beta\mathbb{E}_{q(\mathbf{s}_t|\mathbf{x}_t)}\left[\mathcal{H}(\pi(\cdot|\mathbf{s}_t))\right]\right)\right]. \tag{26}$$

The conditional entropy $\mathcal{H}(\pi(\cdot|\mathbf{s}_t))$ represents the entropy around a single mode. By *minimizing* this in conjunction with maximizing the marginal entropy, we incentivize the policy to disperse its density to other modes, encouraging it to explore and find all regions with high reward. This allows the latent variable policy to make better use of its source of randomness $\mathbf{s}_t$, and encourages a nonlinear mapping between $\mathbf{s}_t$ and the action space. This is in stark contrast to entropy maximization with a Gaussian policy, where random noise is simply added linearly and independently to the action. This technique is only useful for a few experiments (hopper_hop humanoid_run humanoid_stand quadruped_escape quadruped_run reacher_easy walker_run) on state-based model-free experiments. In the pixel-based experiments where SMAC leverages a world model, the distribution of the latent variable is learned within the world model, thus there is no need to further involve such a regularizer.

**Unadopted techniques** Our proposed technique is universal and could be applied to any MaxEnt RL algorithm. For example, we also try to combine Dreamer with MaxEnt in the policy optimization part, and wish to further improve the performance with our method. Nonetheless, as stated in the Appendix of Hafner et al. (2020), there is no much positive effect of introducing MaxEnt principle into Dreamer implementation. Indeed, we find that doing MaxEnt on the actor of Dreamer will in fact lower down the sample efficiency. Therefore, we think it is not meaningful to put the experiments of this part into this work.

Another unused technique lies in latent variable modeling. MLMC with finite samples (*i.e.*, $K < \infty$) still gives a biased estimator. On the other hand, Russian Roulette estimator (Kahn, 1955) enables an unbiased estimate of an infinite series of summation with the help of randomized truncations, together with corresponding term upweighting operation. This technique is also used in many modern machine learning problems (Xu et al., 2019; Chen et al., 2019). As a result, we also try to introduce Russian Roulette calculation into our MLMC estimator. However, we do not find much evident improvement in our RL experiments, thus we do not take this technique into the final SMAC algorithm.

## C ADDITIONAL DETAILS REGARDING EXPERIMENTS

For all episode return curves, we report the mean over 5 seeds and $95\%$ confidence intervals.

---

**Algorithm 1** SMAC (without a world model)

1: **for** each step **do**
2:     //Env interaction
3:     $\mathbf{a} \sim \pi(\mathbf{a}|\mathbf{s}), \mathbf{s} \sim q(\mathbf{s}|\mathbf{x})$
4:     $r, \mathbf{x}' \leftarrow \texttt{env.step(a)}$
5:     $\mathcal{D} \leftarrow \mathcal{D} \cup \{(\mathbf{x}, \mathbf{a}, r, \mathbf{x}')\}$
6:
7:     //Critic learning
8:     Update $Q$ to minimize Eq. 23
9:
10:     //Actor learning
11:     Calculate $\widetilde{\mathcal{H}}_K^{\text{MLMC}}$ via Eq. 12
12:     Update $\pi$ to minimize Eq. 24
13: **end for**

---

**Algorithm 2** SMAC (with a world model)

1: **for** each step **do**
2:     //Environment interaction
3:     **for** $t = 1 \dots T$ **do**
4:         $\mathbf{a}_t \sim \pi(\mathbf{a}_t|\mathbf{s}_t), \mathbf{s}_t \sim q(\mathbf{s}_t|\mathbf{s}_{t-1}, \mathbf{a}_{t-1}, \mathbf{x}_t)$
5:         $r_t, \mathbf{x}_{t+1} \leftarrow \texttt{env.step(}\mathbf{a}_t\texttt{)}$
6:     **end for**
7:     $\mathcal{D} \leftarrow \mathcal{D} \cup \{(\mathbf{x}_t, \mathbf{a}_t, r_t)_{t=1}^T\}$
8:
9:     //World model learning
10:     Update world model to maximize Eq. 22
11:
12:     //Critic learning
13:     Update $Q$ to minimize Eq. 25
14:
15:     //Actor learning
16:     Calculate $\widetilde{\mathcal{H}}_K^{\text{MLMC}}$ via Eq. 12
17:     Calculate $\widetilde{Q}_K$ via Eq. 14
18:     Update policy to maximize $\widetilde{Q}_K + \alpha \widetilde{\mathcal{H}}_K^{\text{MLMC}}$
19: **end for**

---

**Regarding Figure 3**   We run SMAC on DMC finger spin environment with different marginal log probability estimators. We obtain the ground truth value via expensive Monte Carlo estimation with $1 \times 10^5$ samples. From the figure, we can see that there is little hope of using a naïve upper bound of entropy for MaxEnt RL, where a reasonable scale of the entropy term is $\sim 10^0$. For IWAE (Burda et al., 2016) implementation, we set the number of particles to 32. Other values for the number of particles give similar results.

**Regarding Section 5.1**   For model-free experiments, the agents are fed with state-based inputs. We follow the PyTorch model-free SAC implementation of Tandon (2020) for this part. The actor is parametrized by a tanh-Gaussian distribution. For our method, we additionally use a two layer MLP to parametrize the latent distribution $q(\mathbf{s}|\mathbf{x})$. We set the neural network width of the baselines and SMAC to 400 and 256 respectively to keep comparable number of parameters. For the entropy coefficients, we use the same autotuning approach from SAC (Haarnoja et al., 2018b). We follow Fujimoto et al. (2018) for the TD3 implementation details, except that we do not take its $1 \times 10^{-3}$ learning rate. This is

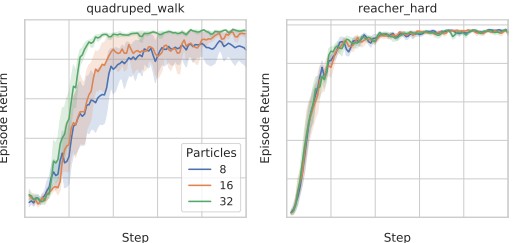

Figure 9: Ablation study of the number of particles for SMAC on `quadruped walk` and `reacher hard` environments. The effect is evident on some but not all environments.

because we found the original learning rate gives very poor performance in our experiments, so instead we set its learning rate to $3 \times 10^{-4}$, which is empirically much better and also consistent with two other algorithms. We conduct ablation study for the number of particles (*i.e.*, $K$ in Equation 12) used in Figure 5. This indicates that the number of level / particles used in the estimation has an effect on some of the environments. We choose the best hyperparameters (number of particles in $\{8, 16, 32\}$, dimension of the latent in $\{8, 16, 32\}$) for each environment. We show the full set of experimental results in Figure 10.

**Regarding other probabilistic policy modeling methods**   We further compare with normalizing flow based policy and mixture-of-experts. For the normalizing flow based method, we follow the practice of Haarnoja et al. (2018a); Ward et al. (2019) to use RealNVP (Dinh et al., 2017) architecture for the policy distribution $\pi(\mathbf{a}|\mathbf{s})$. The neural network is also followed by a tanh transformation

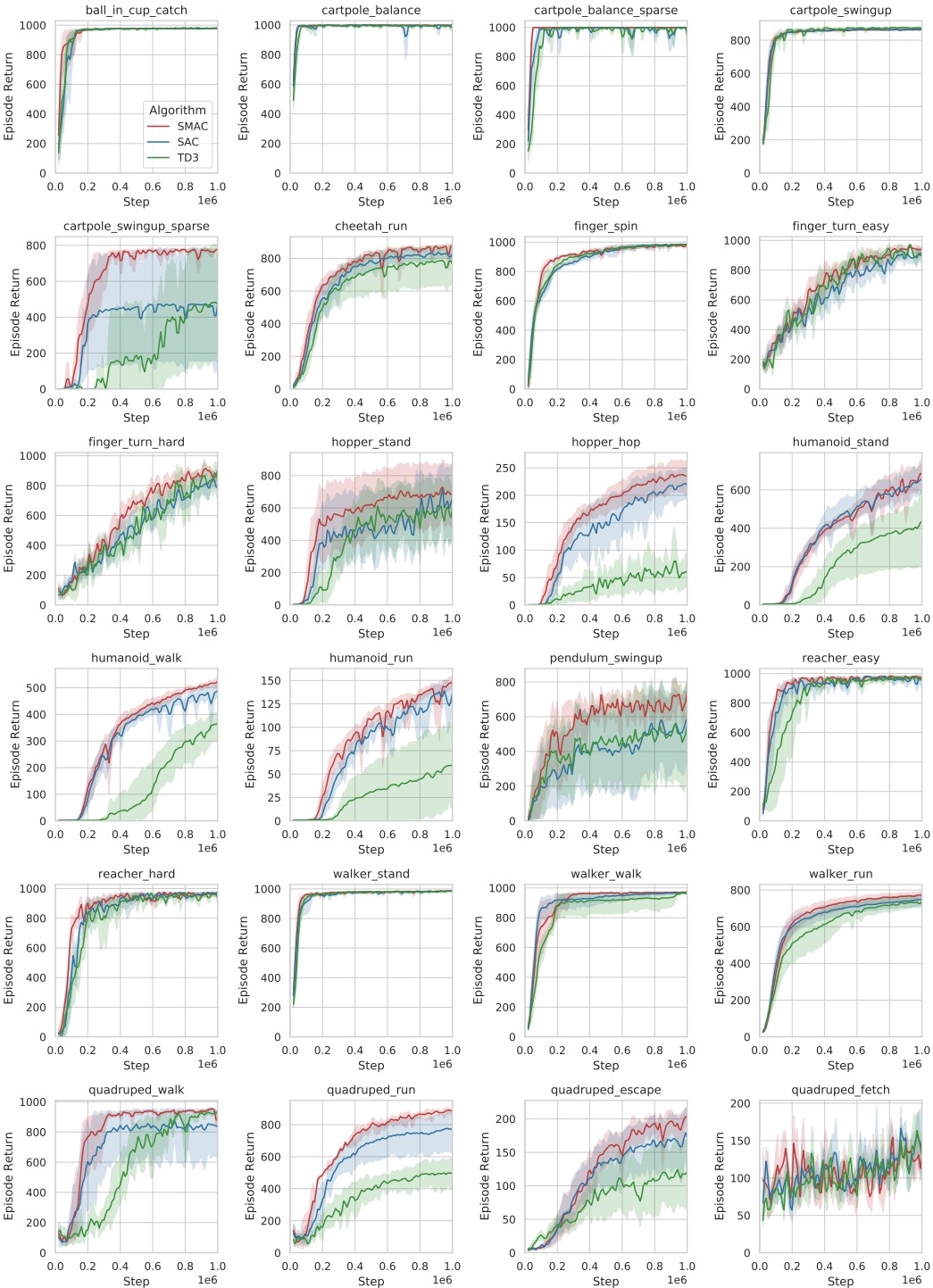

Figure 10: Experiment results on different DMC environments with state-based observations.

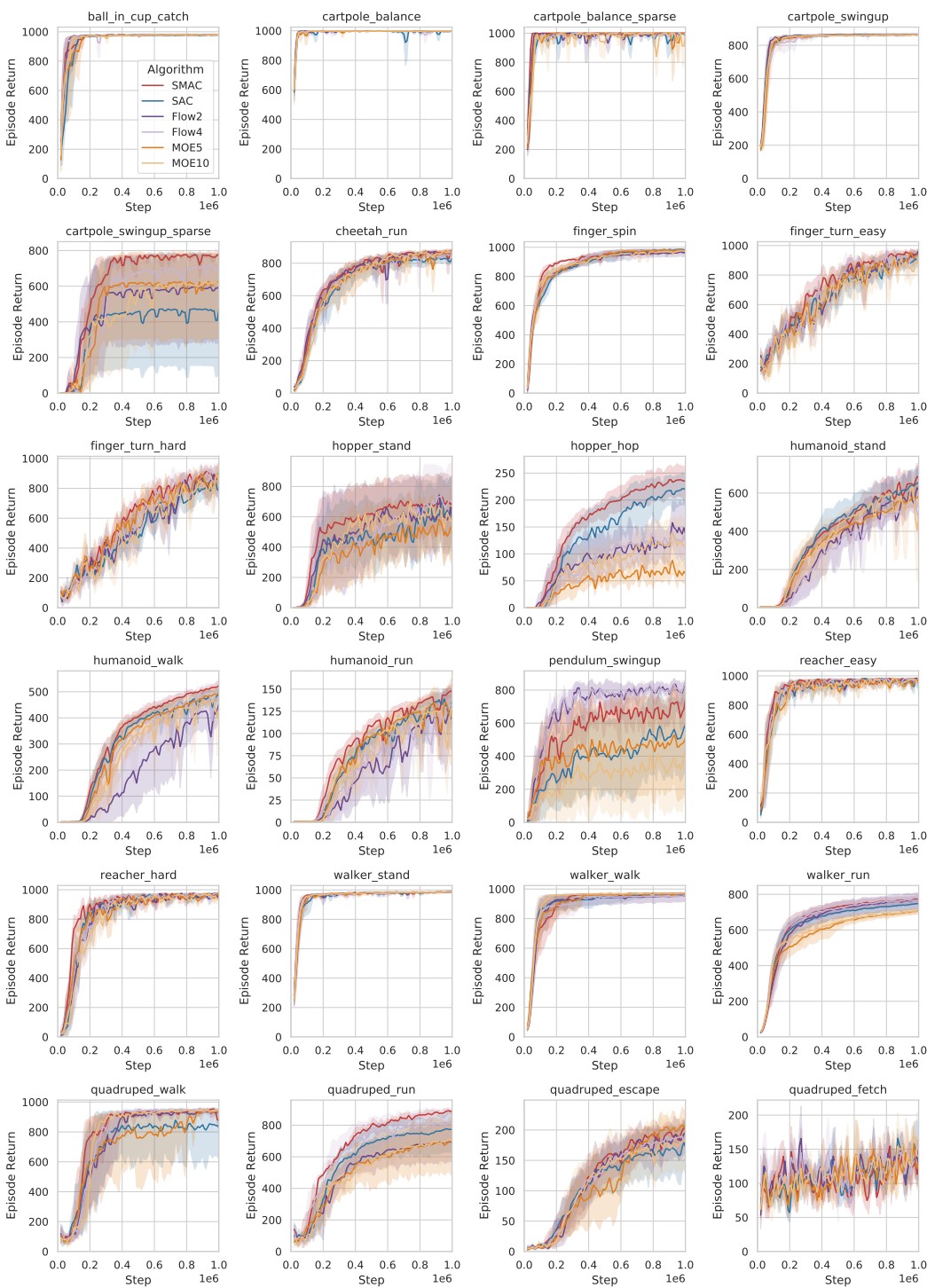

Figure 11: Experiment results about comparison with other probabilistic policy modeling methods on different DMC environments with state-based observations. "Flow2" and "Flow4" refer to normalizing flow based policy with two or four RealNVP blocks as backend, while "MOE5" and "MOE10" refer to probabilistic mixture-of-experts policy with five or ten mixtures. MOE, SAC and SMAC share similar number of parameters, while flow methods have about two or four times more parameters. Our proposed SMAC general achieves the best sample efficiency on a majority of environments.

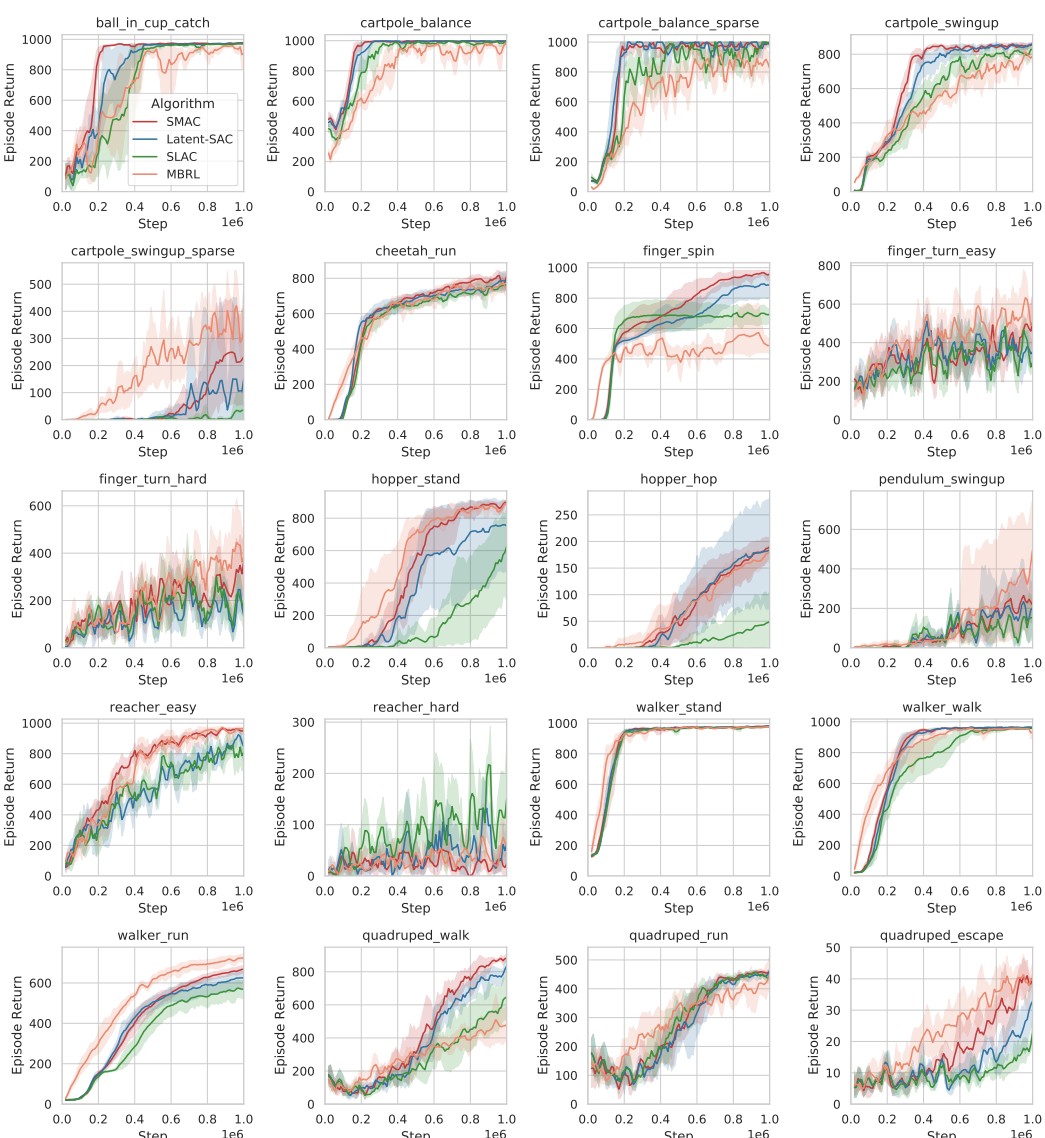

Figure 12: Experiment results on different DMC environments with pixel-based observations.

as other methods. We experiment with two-block and four-block RealNVPs, which makes their number of parameters to be approximately twice and four times as the number of SAC and SMAC methods (these two share approximate the same number of parameters). We show their performance in Figure 11 as "Flow2" and "Flow4", respectively. About the probabilistic mixture-of-experts (MOE) method, we follow the practice of Ren et al. (2021) to use a five Gaussian mixture and a ten Gaussian mixture. We show their performance in Figure 11 as "MOE5" and "MOE10", respectively. This probabilistic MOE shares roughly around the same number of parameters and compute cost as SAC and SMAC. To conclude, SMAC achieves more favorable performance across a majority of the control tasks with less or equal number of parameters (notice that SMAC is the red curve in the figure). Normalizing flow based polices outperform SMAC on only one environment, while MOE requires the use of biased gradient estimators which often required extra hyperparameter tuning or could lead to worse performance.

**Regarding Section 5.2** For this part, the agents are fed with pixel-based inputs. We mainly follow the PyTorch world model implementation of Lin (2022) for this part. We implement the Latent-SAC algorithm according to instructions and hyperparameters in Wang et al. (2022). We implement the SLAC (Lee et al., 2020a) algorithm following its original github repo (Lee et al., 2020b). Note that Latent-SAC and SLAC are two different algorithms, in the sense of actor modeling and world model design, although they both build on a SAC backend. We select the hyperparameters in the same way as the above part. We show the full version of experimental results in Figure 12. We do not plot the results of the `humanoid` domain as well as `quadruped fetch`, as all methods could not obtain meaningful results within 1 million frames.

For the robustness experiments, we add two kinds of noises in the pixel space, namely Gaussian perturbation and sensor missing perturbation. For Gaussian perturbation, we add isotropic Gaussian noise with scale $0.01$ and $0.05$. For sensor missing, we randomly drop the pixel value for each dimension to zero according to a Bernoulli distribution (with parameter $0.01$ and $0.05$) in an independent way. For results in Table 1, we report the best episodic reward across training iterations with standard deviation estimated from 5 seeds.

# D  THEORETICAL DERIVATIONS

## D.1  UNIVERSALITY OF LATENT VARIABLE MODELS

We would first need the help of the following Lemma for the proof.

**Lemma 2.** *For any continuous $d$-dimensional distribution $p^*(x)$ and $\forall \epsilon > 0$, there exists a neural network $\Psi : \mathbb{R} \to \mathbb{R}^d$ with finite depth and width that satisfies $\mathcal{W}\left[\Psi \# \mathcal{N}(0,1) || p^*(\cdot)\right] \leq \epsilon$. Here $\#$ is the push-forward operator, $\mathcal{W}(\mu, \nu) := \inf_{\pi \in \Pi(\mu, \nu)} \int |x - y| \, d\pi(x, y)$ is the Wasserstein metric, and $\mathcal{N}(0,1)$ is the standard Gaussian distribution.*

*Proof.* The Theorem 5.1 of Perekrestenko et al. (2020) shows that for any $p^*(x)$ and $\epsilon > 0$, there exists a nonlinear ReLU neural network with finite size $\tilde{\Psi} : \mathbb{R} \to \mathbb{R}^d$ that satisfies $\mathcal{W}\left[\tilde{\Psi} \# U[0,1] || p^*(\cdot)\right] \leq \epsilon$, where $U[0,1]$ is the uniform distribution on $[0,1]$. On the other hand, it is well known that the cumulative distribution function (cdf) of standard Gaussian $\Phi(\cdot) : \mathbb{R} \to \mathbb{R}$ could map the standard Gaussian to the uniform distribution $U[0,1]$, thus we have the construction of $\Psi := \tilde{\Psi} \circ \Phi$. $\square$

We use this result to prove Proposition 1.

**Proposition 1.** *For any $d$-dimensional continuous distribution $p^*(x)$, there exist a sequence of two-level latent variable model $p_n(x) = \int p_n(x|z)p_n(z) \, dz, n \in \mathbb{N}_+$ that converge to it, where both $p_n(x|z)$ and $p_n(z)$ are factorized Gaussian distributions with mean and variance parameterized by neural networks.*

*Proof of Proposition 1.* We let $z \in \mathbb{R}$ and set $p_n(z) = \mathcal{N}(0,1), \forall n \in \mathbb{N}_+$. From the above Lemma 2, we know that $\forall n \in \mathbb{N}_+, \exists \Psi_n : \mathbb{R} \to \mathbb{R}^d$ s.t. $\mathcal{W}\left[\Psi_n \# p_n(z) || p^*(x)\right] \leq \frac{1}{n}$, where $\Psi_n$ is a finite size

neural network. We then set $p_n(x|z) = \mathcal{N}(x; \Psi_n(z), \frac{1}{n^2})$. Note that this falls into the category of factorized Gaussian.

Let $\pi_0$ be a coupling between $p_n(x)$ and $\Psi_n \# p_n(z)$, where $\pi_0(x, x')$ is the joint distribution over $(x, x')$ and $x' = x + \zeta/n$, $\zeta \sim \mathcal{N}(0, 1)$, $x \sim \Psi_n \# p_n(z)$. We thus have $\mathcal{W}[p_n(x)||\Psi_n \# p_n(z)] \leq \int |x - x'| \, d\pi_0(x, x') = \frac{1}{\sqrt{2\pi}n} < \frac{1}{n}$. Since the Wasserstein metric satisfies the triangle inequality, we have $\mathcal{W}[p_n(x)||p^*(x)] \leq \mathcal{W}[p_n(x)||\Psi_n \# p_n(z)] + \mathcal{W}[\Psi_n \# p_n(z)||p^*(x)] \leq \frac{2}{n} \overset{n \to \infty}{\longrightarrow} 0$. $\qquad\square$

### D.2 TIGHT LOWER BOUND ON THE MARGINAL ENTROPY

**Proposition 3** (Lower bound of marginal entropy). *For a latent variable policy $\pi(\mathbf{a}|\mathbf{h}) := \int \pi(\mathbf{a}|\mathbf{s})q(\mathbf{s}|\mathbf{h}) \, dz$ with prior $q(\mathbf{s}|\mathbf{h})$ and likelihood $\pi(\mathbf{a}|\mathbf{s})$, consider*

$$\widetilde{\mathcal{H}}_K(\mathbf{h}) \triangleq \mathbb{E}_{\mathbf{a} \sim \pi(\mathbf{a}|\mathbf{h})} \mathbb{E}_{\mathbf{s}^{(0)} \sim p(\mathbf{s}|\mathbf{a},\mathbf{h})} \mathbb{E}_{\mathbf{s}^{(1:K)} \sim q(\mathbf{s}|\mathbf{h})} \left[ -\log \left( \frac{1}{K+1} \sum_{k=0}^{K} \pi\left(\mathbf{a}|\mathbf{s}^{(k)}\right) \right) \right]. \qquad (27)$$

*where $p(\mathbf{s}|\mathbf{a}, \mathbf{h}) \propto \pi(\mathbf{a}|\mathbf{s})q(\mathbf{s}|\mathbf{h})$ is the posterior and $K$ is any positive integer, then the following holds:*

*(1)* $\widetilde{\mathcal{H}}_K(\mathbf{h}) \leq \mathcal{H}(\pi(\cdot|\mathbf{h})) \triangleq -\int_{\mathcal{A}} \log \int_{\mathcal{S}} \pi(\mathbf{a}|\mathbf{s})q(\mathbf{s}|\mathbf{h}) \, d\mathbf{s} \, d\mathbf{a}$,

*(2)* $\widetilde{\mathcal{H}}_K(\mathbf{h}) \leq \widetilde{\mathcal{H}}_{K+1}(\mathbf{h})$,

*(3)* $\lim_{K \to \infty} \widetilde{\mathcal{H}}_K(\mathbf{h}) = \mathcal{H}(\pi(\cdot|\mathbf{h}))$.

The following proofs roughly follow the derivations from Sobolev & Vetrov (2019). We describe them here for completeness.

For (1):

*Proof.* We write

$$\mathcal{H}(\pi(\cdot|\mathbf{h})) - \widetilde{\mathcal{H}}_K(\mathbf{h}) = \mathbb{E}_{\mathbf{s}_0 \sim p(\mathbf{s}|\mathbf{a},\mathbf{h})} \mathbb{E}_{\mathbf{s}^{(1:K)} \sim q(\mathbf{s}|\mathbf{h})} \left[ \log \left( \frac{1}{K+1} \sum_{k=0}^{K} \frac{p(\mathbf{s}^{(k)}|\mathbf{a}, \mathbf{h})}{q(\mathbf{s}^{(k)}|\mathbf{h})} \right) \right]$$

$$\triangleq \mathbb{E}_{\mathbf{s}_0 \sim p(\mathbf{s}|\mathbf{a},\mathbf{h})} \mathbb{E}_{\mathbf{s}^{(1:K)} \sim q(\mathbf{s}|\mathbf{h})} \left[ \log \frac{p(\mathbf{s}_0|\mathbf{a}, \mathbf{h})q(\mathbf{s}^{(1:K)}|\mathbf{h})}{w(\mathbf{s}^{(0:K)}|\mathbf{a}, \mathbf{h})} \right]$$

$$= D_{\mathrm{KL}} \left[ p(\mathbf{s}_0|\mathbf{a}, \mathbf{h})q(\mathbf{s}^{(1:K)}|\mathbf{h})||w(\mathbf{s}^{(0:K)}|\mathbf{a}, \mathbf{h}) \right],$$

where $w(\mathbf{s}^{(0:K)}|\mathbf{a}, \mathbf{h}) = \frac{p(\mathbf{s}^{(0)}|\mathbf{a},\mathbf{h})q(\mathbf{s}^{(1:K)}|\mathbf{h})}{\frac{1}{K+1}\sum_{k=0}^{K} \frac{p(\mathbf{s}^{(k)}|\mathbf{a},\mathbf{h})}{q(\mathbf{s}^{(k)}|\mathbf{h})}}$. We only need to show that $w(\mathbf{s}^{(0:K)}|\mathbf{a}, \mathbf{h})$ is a normalized density function.

Consider such generation process:

1. sample $K + 1$ samples $\tilde{\mathbf{s}}^{(k)} \sim q(\mathbf{s}|\mathbf{h}), k = 0, \dots, K$,

2. set weight for each sample $w_k = \pi(\mathbf{a}|\tilde{\mathbf{s}}^{(k)})$,

3. sample a categorical random variable $h \sim Cat\left(\frac{w_0}{\sum_{k=0}^{K} w_k}, \dots, \frac{w_K}{\sum_{k=0}^{K} w_k}\right)$,

4. put the $h$-th sample to the first: $\mathbf{s}_0 = \tilde{\mathbf{s}}^{(h)}, \mathbf{s}^{(1:K)} = \tilde{\mathbf{s}}^{(\backslash h)}$.

It is easy to see the joint probability of this generation process is

$$p(\tilde{\mathbf{s}}^{(0:K)}, \mathbf{s}^{(0:K)}, h) = q(\mathbf{s}^{(0:K)}|\mathbf{h}) \frac{w_h}{\sum_{k=0}^{K} w_k} \delta(\mathbf{s}_0 - \tilde{\mathbf{s}}^{(h)}) \delta(\mathbf{s}^{(1:K)} - \tilde{\mathbf{s}}^{(\backslash h)}).$$

Then the marginal of $\mathbf{s}^{(0:K)}$ is

$$
\int \sum_{h=0}^{K} q(\mathbf{s}^{(0:K)}|\mathbf{h}) \frac{w_h}{\sum_{k=0}^{K} w_k} \delta(\mathbf{s}_0 - \tilde{\mathbf{s}}^{(h)}) \delta(\mathbf{s}^{(1:K)} - \tilde{\mathbf{s}}^{(\backslash h)}) \, d\tilde{\mathbf{s}}^{(0:K)}
$$

$$
= (K+1) \int q(\mathbf{s}^{(0:K)}|\mathbf{h}) \frac{w_0}{\sum_{k=0}^{K} w_k} \delta(\mathbf{s}^{(0)} - \tilde{\mathbf{s}}^{(0)}) \delta(\mathbf{s}^{(1:K)} - \tilde{\mathbf{s}}^{(1:K)}) \, d\tilde{\mathbf{s}}^{(0:K)}
$$

$$
= (K+1) \frac{q(\mathbf{s}^{(0:K)}|\mathbf{h}) \pi(\mathbf{a}|\mathbf{s}^{(0)})}{\sum_{k=0}^{K} \pi(\mathbf{a}|\mathbf{s}^{(k)})} = \frac{q(\mathbf{s}^{(1:K)}|\mathbf{h}) p(\mathbf{s}^{(0)}|\mathbf{a},\mathbf{h})}{\frac{1}{K+1} \sum_{k=0}^{K} \frac{p(\mathbf{s}^{(k)}|\mathbf{a},\mathbf{h})}{q(\mathbf{s}^{(k)}|\mathbf{h})}} = w(\mathbf{s}^{(0:K)}|\mathbf{a},\mathbf{h}).
$$

Thus $w(\mathbf{s}^{(0:K)}|\mathbf{a},\mathbf{h})$ is a normalized density function.

$\square$

For (2).

*Proof.* We write

$$
\widetilde{\mathcal{H}}_{K+1}(\mathbf{h}) - \widetilde{\mathcal{H}}_K(\mathbf{h}) = \mathbb{E}_{\mathbf{s}_0 \sim p(\mathbf{s}|\mathbf{a},\mathbf{h})} \mathbb{E}_{z_{1:K+1} \sim q(\mathbf{s}|\mathbf{h})} \left[ \log \left( \frac{\frac{1}{K+1} \sum_{k=0}^{K} \pi(\mathbf{a}|\mathbf{s}^{(k)})}{\frac{1}{K+2} \sum_{k=0}^{K+1} \pi(\mathbf{a}|\mathbf{s}^{(k)})} \right) \right]
$$

$$
\triangleq \mathbb{E}_{\mathbf{s}_0 \sim p(\mathbf{s}|\mathbf{a},\mathbf{h})} \mathbb{E}_{z_{1:K+1} \sim q(\mathbf{s}|\mathbf{h})} \left[ \log \frac{p(\mathbf{s}^{(0)}|\mathbf{a},\mathbf{h}) q(\mathbf{s}^{(1:K+1)}|\mathbf{h})}{w(\mathbf{s}^{(0:K+1)}|\mathbf{a},\mathbf{h})} \right]
$$

$$
= D_{\mathrm{KL}} \left[ p(\mathbf{s}^{(0)}|\mathbf{a},\mathbf{h}) q(\mathbf{s}^{(1:K+1)}|\mathbf{h}) \| v(\mathbf{s}^{(0:K+1)}|\mathbf{a},\mathbf{h}) \right],
$$

where $v(\mathbf{s}^{(0:K+1)}|\mathbf{a},\mathbf{h}) = p(\mathbf{s}^{(0)}|\mathbf{a},\mathbf{h}) q(\mathbf{s}^{(1:K+1)}|\mathbf{h}) \frac{\frac{1}{K+2} \sum_{k=0}^{K+1} \pi(\mathbf{a}|\mathbf{s}^{(k)})}{\frac{1}{K+1} \sum_{k=0}^{K} \pi(\mathbf{a}|\mathbf{s}^{(k)})}$. We could then show that $v(\mathbf{s}^{(0:K+1)}|\mathbf{a},\mathbf{h})$ is a normalized density function similarly as (1). $\square$

For (3).

*Proof.* For the estimator, we have

$$
\frac{1}{K+1} \sum_{k=0}^{K} \pi(\mathbf{a}|\mathbf{s}^{(k)}) = \overbrace{\frac{1}{K+1} \pi(\mathbf{a}|\mathbf{s}^{(0)})}^{A_K} + \overbrace{\frac{K}{K+1}}^{B_K} \overbrace{\frac{1}{K} \sum_{k=1}^{K} \pi(\mathbf{a}|\mathbf{s}^{(k)})}^{C_K}
$$

By law of large numbers, we have $A_K \to 0, B_K \to 1, C_K \to p(x)$. Thus the limit of the left-hand side is $p(x)$. $\square$

