# OpenReview forum: "Latent State Marginalization as a Low-cost Approach for Improving Exploration"
_ICLR.cc/2023/Conference — ICLR 2023 poster_

### Official Review · Reviewer_fjCR · 2022-10-20

**Confidence:** 4
**Correctness:** 3
**Technical Novelty And Significance:** 2
**Empirical Novelty And Significance:** 2
**Recommendation:** 6

**Clarity, Quality, Novelty And Reproducibility:**

**Clarity** — The clarity of the paper, while generally good, could be somewhat improved. One unclear aspect was the notation. For instance, the authors conflate the latent state of the environment with the latent state of the model, when these two variables need not be defined in the same manner. In Eq. 9, the authors seeming write the entropy as being over the history (rather than the latent state and action), while in Eq. 10, they write the entropy as being conditional on the history. (Note: in Eq. 9, there’s also a typo (?) in which r is denoted as a distribution over the latent state). And in Eq. 11, the authors refer to $\pi$ as both the action policy as well as the posterior over latent states.

I found Section 3.2.1. somewhat difficult to follow, as it’s unclear exactly where the authors are employing amortized variational inference vs. exact probabilistic inference, etc. That is, it’s not clear how many additional models are required to implement the authors’ approach. In a similar vein, the authors claim to use “standard variational inference,” when, as far as I can tell, they are making an assumption of amortization, i.e., I doubt that they are optimizing $q$ or $\pi$ on a per-state basis.

In some cases, the underlying model assumptions go unstated, making it difficult to understand where the authors are making deliberate choices vs. following convention. To give some examples, the authors employ a filtering variational posterior over the world model latent state, yielding a looser bound than a smoothing variational posterior, without justifying this choice. They also insist on conditioning the policy purely on the current latent state, rather than (correctly) allowing the policy to depend on the entire history.

**Quality** — Overall, the quality of the paper is reasonable, but there are several aspects that could be improved. In some places, the authors use sweeping generalizations or mischaracterizations.

- For instance, on the first page, the authors claim that they want to learn complex, multi-modal behaviors, whereas many existing algorithms rely on local perturbations around a singe action. First, even with a uni-modal policy mapping, compounding local perturbations across time will still yield multi-modal trajectory densities. Second, this restriction to local perturbations is, in fact, a result of using parametric policies (i.e., neural networks), whereas an iteratively-optimized Gaussian policy (e.g., optimized through planning) can recover multiple modes. Finally, the authors do not definitively demonstrate that their approach results in complex multi-modal behavior.

- The authors state that “a majority of approaches for handling partial observability make use of world models,” which is not well-supported by the literature. A variety of existing works do not use latent variable models to handle partial observability, instead relying on recurrent networks (see, e.g., MuZero/Muesli) to integrate information.

- Elsewhere in the paper, the authors claim that “SAC is often restricted to the use of policies where the entropy can be computed efficiently, e.g. a factorized Gaussian policy…” However, in the case of SAC, the authors employ a tanh on the policy output, using only a sample-based estimate of the entropy. Likewise, in many KL-regularized RL methods, e.g., MPO (Abdolmaleki, et al., 2018), the authors resort to sample-based estimates.

- The authors claim, without any rigorous justification, that “with standard neural network architectures, a latent variable policy can universally approximate any distribution if given sufficient capacity.” This is merely presented as a “proposition,” without any follow-up analysis to determine the degree to which this holds in practice, e.g., does the KL between the policy and Boltzmann optimal policy not decrease with added latent variables? If so, this would seem to contradict the finding from the hierarchical VAE and diffusion community, which is that hierarchy depth improves distribution flexibility. Is this purely a function of the “capacity” of the policy networks, or are there other considerations?

- The authors claim that “latent variables can be used within the $Q$-function to better aggregate uncertainty” but they do not verify this empirically.

As mentioned previously, the baselines for empirical comparison do not isolate the factors of the authors’ proposal (i.e., hierarchy, multiple samples, variance reduction, etc.). This makes it difficult to assess the origin of the performance benefits, diminishing the quality of the empirical analysis.

Minor point: Figure 5 is missing axis numbers.

**Novelty** — The novelty of the proposed approach is somewhat limited. The main contribution of the paper is in combining several previously proposed ideas to improve multi-sample entropy and value estimates in the case of latent variable policies. This is entirely within existing framings, i.e., SAC and world models. And the specific combination and application of these techniques is novel to RL, it’s fairly intuitive that using additional samples to improve entropy and values estimates should yield improved performance. In this sense, I found the proposed approach to be only marginally novel.

**Reproducibility** — While it’s difficult to assess, my guess is that the results are reproducible. The performance of the baselines looks to be in-line with previous works, suggesting that the authors’ implementation is reasonable. And there is a clear trend of SMAC being marginally better than previous methods across environments (with multiple seeds), demonstrating that the improved estimator likely does improve performance. I appreciate that the authors included some ablations to assess the effect of different hyperparameters.

**Strength And Weaknesses:**

**Strengths**

- **Proposes a principled approach to improving entropy and Q-value estimates.** To the best of my understanding, the method proposed here appears principled, building off of previous methods for entropy estimation. While many previous works have generally ignored the fact that world model latent variables induce hierarchical policies, few previous works have taken advantage of this fact to improve entropy and value estimators.

- **Builds off of popular existing methods.** The paper builds off of several established algorithms in the literature, namely SAC and world models. These algorithms have fairly widespread adoption in the academic literature, so there’s a higher likelihood that methods built upon these algorithms will be more widely adopted, particularly if they are not difficult to implement.

- **Uses standard benchmark setups/environments.** The authors evaluate SMAC using the DeepMind control suite, a standard set of continuous control environments built on MuJoCo. These environments are fairly commonly used in the literature, so demonstrating results here will likely be compelling to the deep RL sub-field.

**Weaknesses**

- **Fairly complex method for a fairly minimal performance boost.** Early in the paper, the authors lament that “the use of more expressive models have not gained nearly as much traction in the community…these constructions often result in complicate[d] training procedures and are inefficient in practice.” Two thoughts here: First, I would disagree with this characterization and analysis. Normalizing flow-based policies do not yield a significantly more complicated or inefficient procedure. Indeed, many modern neural network libraries (PyTorch, Tensorflow, and JAX/Distrax) make it trivial to implement these policies and sample / evaluate log-densities. It’s not that these more expressive policies aren’t useful. Rather, the limited boost in performance in our fairly simple benchmark environments hasn’t sufficiently justified their widespread adoption. Second, the method presented here strikes me as more complicated than, for example, normalizing flow-based policies. I find it difficult to believe that SMAC would receive widespread adoption when normalizing flow-based policies have not.

- **Unclear if the baselines are really appropriate/fair comparisons.** The authors, at times, appear to confuse the distinction between several proposals: 1) using hierarchical policies, 2) using more sophisticated/multi-sample estimators, and 3) their particular approach for estimation, which itself consists of multiple components (entropy and Q-value estimators). For instance, in some of their experiments, the authors only compare with SAC and TD3, claiming to show the benefits of hierarchical policies. This is hardly a novel finding, as it’s well-known that more expressive policy distributions generally outperform less expressive ones. At the very least, I would have expected comparisons with normalizing-flow-based policies, which, having personally implemented these in one afternoon, are not difficult to use (and these do, indeed, outperform Gaussian policies in SAC). Likewise, in the only other set of experiments, the authors compare with “Latent-SAC”, SLAC, and Dreamer, but it’s not at all clear what these baseline comparisons are meant to accomplish. Is the boost in performance coming from using multiple samples, or is it the estimator itself? Is it from having better value estimates, entropy estimates, or do they both contribute? Likewise, as far as I understand, this estimator could also be used with a model-based value estimator, so it’s unclear whether the comparison with Dreamer tells us much about the setup proposed here.

- **Somewhat minimal contribution.** While I’m not discounting the insight and effort required to develop the main techniques (nested estimator with MLMC) in the paper, the main contribution of the paper reduces to just drawing more samples for estimation. Assuming we have a reasonable estimator, it’s hardly surprising that drawing additional samples should yield improved entropy and value estimates, and likely, improved performance. The proposal of the paper is effectively to use a well-known knob within the existing RL framework to improve performance, which I view as a somewhat minimal contribution.

**Summary Of The Paper:**

The paper proposes to marginalize (or rather, draw multiple samples from) latent variable policies as a means of improving estimates of policy entropy and Q-value estimates. The authors point out that a naïve ELBO estimate can result in a loose bound (i.e., a biased estimate), so they suggest using a nested estimator and multi-level Monte Carlo. They perform experiments using continuous control environments from DeepMind control suite, both in the setting of joint-based and pixel-based observations. They find that their method outperforms SAC and TD3 in the joint-based setting and outperforms various world-model-based setups in the pixel-based setting.

**Summary Of The Review:**

The paper provides a technique for improving entropy and values estimates in latent variable policies using multiple samples. While I believe that this is a useful idea, the somewhat limited novelty, combined with the marginal performance boost (over inappropriate baselines) leads me to feel that this paper is somewhat below the acceptance threshold. The paper could be improved by improving the baselines to illuminate the various aspects (i.e., hierarchy vs. multi-sample vs. variance) of the proposed method. The authors may also consider whether this method allows one to scale up to larger hierarchical policies to tackle more challenging tasks where performance gains may be more significant.


**UPDATE:**
I appreciate that the authors have corrected various issues in the text that I raised. The normalizing flows experiments also add a useful multi-modal baseline to the results. I have increased my score from 5 --> 6 to acknowledge the effort on the part of the authors. However, I still feel that the complexity of the method and its limited novelty do not warrant the somewhat minimal boost in performance as currently demonstrated in the paper. For instance, in light of the normalizing flows experiments (Figure 11), it appears SMAC has statistically significant performance benefits on only 1 (or maybe 2) of the 24 DM control suite environments. I do not find this to be a compelling argument in favor of using SMAC.

---

> ### Author Response · Authors · 2022-11-10
> **Response to Reviewer fjCR (1)**
>
> Thanks for the insightful review! There seem to be some misunderstanding as a result of our negligence in writing, and we hope our explanation below could help you address them. For better demonstration, we group related questions together.
>
> **About comparison with other probabilistic modeling methods.**  We agree that normalizing flow (NF) is an important baseline that we should compare with. We have added comparisons to normalizing flow policies in Fig. 11. Furthermore, we follow the practice of [1] to use two-layer and four-layer RealNVP normalizing flow respectively to replace the original Gaussian policy network. A tanh transformation is also used. The number of parameters of such methods is twice and four times as the original SAC and SMAC method (which shares the same number of parameters) and the training speed is already 1.5 ~ 2 times slower. Our SMAC method could outperform both SAC and NF-based SAC algorithms on most of the environments. We found that NF policies required more tuning and can be unreliable if the architecture is not chosen correctly. The invertible transformation is unconstrained, making its optimization landscape to be hard with weird learning dynamics. Whereas our approach has the same number of parameters as a SAC baseline and only requires making a layer stochastic.
> We have also included a mixture-of-experts based policy as another baseline. As this involves a discrete latent variable, we had to resort to biased gradient estimators for training. See Appendix C for more details.
>
> [1] Patrick Nadeem Ward, et al. Improving exploration in soft-actor critic with normalizing flows policies.

---

> > ### Author Response · Authors · 2022-11-10
> > **Response to Reviewer fjCR (2)**
> >
> > **About the entropy estimator, and its novelty.** There seems to be a misunderstanding here, and thus we want to point out that **our estimation is NOT simply using more particles**. As we have shown in Fig. 3, simply using more particles (i.e., IWAE in the figure) to estimate the entropy with variational bound is numerically unstable. This is because commonly adopted variational bound will give an upper bound of entropy, and **maximizing an upper bound is catastrophic** in MaxEnt RL. That said, ELBO and IWAE (with finite particles) are not only loose bounds, but also exhibit biases in the wrong direction. IWAE estimator will become tighter with more particles, however provided it is still from a wrong direction, maximizing it will lead to numerical issues as can be easily observed when tested. On the other hand, we propose to use a lower bound for entropy estimation in Sec. 3.2.1. The nested estimator in Eq. 11 is effectively a lower bound of latent variable model entropy (whose validity is justified with delicate design, with details shown in Sec. D.2), and is tight when the number of particles goes to infinity. We acknowledge that we are not the first to investigate an entropy lower bound from a probabilistic inference perspective, but our novelty lies in that fact that we are the first work to use latent variable policy effectively in MaxEnt RL community. We advocate such an approach due to its simplicity and low compute cost.
> >
> > **About distinction from different proposals.**  We distinguish the contribution of different aspects to the proposed method as follows: (1) regarding hierarchy, we show that latent variable policy is more favorable than (squash) factorized Gaussian in Fig. 4 in Sec. 5.1 and Fig. 10 in Appendix C; note that our finding shows that the latent variable policy achieves better sample efficiency under same number of parameters. (2) regarding multi-particle estimator and MLMC technique, we provide ablation in Fig. 5 in Sec. 5.1 and Fig. 9 in Appendix C. Importantly, simply using more particles does not improve performance and results in similar training curves as SAC on many environments. This is because using more samples naively creates a worse signal-to-noise ratio, resulting in variance exploding when the number of samples increases (Sec 3.2.2). As a result, the advantage of SMAC is a joint effect from multiple different aspects.
> >
> > **About algorithms in Sec. 5.2 experiments.**   We compare our proposed SMAC against three algorithms in Sec. 5.2, namely Latent-SAC, SLAC and MBRL (i.e., Dreamer). We make the following remarks. (1) Latent-SAC is the main baseline which we would like to compare with. It shares the same architecture with SMAC, while effectively using a poorer entropy estimator (though still a lower bound). Comparison with Latent-SAC effectively corroborates the importance of effective latent marginalization. (2) The MBRL (Dreamer) method uses a similar architecture with our method (SMAC uses a Q function, while MBRL uses a value function estimated from rollouts). In theory, MBRL should be better, since it consumes much more computation to do planning. However, we can see on the majority of the environments our SMAC achieves better performance than the MBRL baseline, which further validates the advantage of our model-free approach. (3) Finally, we include SLAC method, which shares a similar idea with Latent-SAC but with a different world model and choice of architecture (refer to their paper for details).
> >
> > **About using with a model-based value estimator.**  If we understand it correctly, the reviewer is asking about the possibility of using our latent state marginalization technique onto model-based RL methods such as Dreamer. Dreamer itself is not a MaxEnt RL algorithm, and thus there is no need for using our entropy estimation method. We have indeed tried to augment Dreamer with a MaxEnt regularization term but could hardly found any improvement (this is actually also briefly mentioned in Appendix A of Dreamer paper). Since SMAC specifically takes advantage of the MaxEnt framework for exploration, we do not expect to see gains when applied to Dreamer. Moreover, as DreamerV2 is a MaxEnt RL method, we think it is actually possible to deploy SMAC upon it. While the approaches underlying SMAC can be adopted, this requires additional / alternative approaches to handling discrete latent variables, but we agree this is definitely an important future direction of this work.

---

> > > ### Author Response · Authors · 2022-11-10
> > > **Response to Reviewer fjCR (3)**
> > >
> > > ## Clarity responses
> > >
> > > **About notations in Equation 9, 10, 11.**  Thanks for pointing these out! The $r$ in Eq. 9 is a typo and it should be $\tilde{q}$ which denotes the variational distribution. The entropy term of latent variable policy $\pi(\cdot|h_t)$ is conditioning on the $h_t$ as shown in Eq. 8, while the notations in Eq. 9, 10 and 11 mean they are estimators which are functions of $h_t$. For partially observed cases, $h_t$ is the history, while in fully observed Markovian settings, conditioning on $h_t$ is equivalent to conditioning on $x_t$, as discussed in Sec. 3.1.1. For Eq. 11, we used $\pi(s|a, h)$ to denote the true posterior, which is proportional to $\pi(a|s)q(s|h)$. We agree this could be confusing as $\pi$ usually denote a distribution over action. Therefore, we use $p(s|a, h)$ to denote the exact posterior instead in the revision.
> > >
> > > **About amortization.**  We are sorry for the confusion in writing in Sec. 3.2.1. Yes, we are using amortized inference. We already make this clear in the revision version. We have neural networks that output the parameters of the distribution conditional on $h_t$.
> > >
> > > **About additional models.**  The reviewer asks how many additional models are required. The answer is, **we use zero additional modeling in the sense of number of model parameters**. For the experiments in Sec. 5.1, SMAC’s policy contains two MLPs while our implementation of SAC’s policy has one MLP that keeps the same number of parameters as discussed in Appendix C. For Sec. 5.2, both SMAC and latent-SAC use the same architecture, since we use the world model to model $q(s|h)$. What’s more, our method costs almost the same computation time compared to Latent-SAC on realistic visual control tasks. We did experiment using an additional amortized variational posterior for entropy estimation, but preliminary experiments showed that the improvement is not too significant, so we decided to only show the simpler version of SMAC without an additional variational posterior.
> > >
> > > **About filtering and smoothing.**  Sorry about the confusion. Our method does not relate to any smoothing, as it would require a conditioning on future observation information. This is not practical, because we need to make decisions based on the past during evaluation time. Since decision-making is a causal process in the direction of time arrows, we only discuss filtering approaches.
> > >
> > > **About latent state and history.**  The reviewer comments we “conditioning the policy purely on the current latent state, rather than (correctly) allowing the policy to depend on the entire history.” We clarify that for partially observed setup, the current latent $s_t$ is actually conditional on history $h_t$ and we use an inference network  $q(s_t|h_t)$ to model this conditioning. To put it in another way, our policy is indeed conditioning on history. This is discussed precisely in Eq 7, and we refer to Fig. 2 and Fig. 8 for a better visualization of such conditioning. This is also the treatment of Dreamer and several other works based on recurrent state-space model (RSSM), such as PlaNet and DreamerV2.

---

> > > > ### Author Response · Authors · 2022-11-10
> > > > **Response to Reviewer fjCR (4)**
> > > >
> > > > ## Quality responses
> > > >
> > > > Thanks for pointing out these confusing parts in writing quality, and we have already refined them in the revision version.
> > > >
> > > > **About multi-modality.**  We agree with you that iterative optimization method such as cross-entropy method are also capable of finding multiple modes. We point out that its shortcoming is that typically planning with a world model is computational expensive, and we do compare with planning-based method in experimental part (“Dreamer” in Fig. 6 and Fig. 11). Not only that, but we also present a simple bandit setup in Sec. B.1 to demonstrate our approach can recover multi-modal behaviors.
> > > >
> > > > **About handling partial observability.**   We agree on the point that many existing works do not use stochastic latent variable modeling to handle partial observability. For example, at the last sentence of Sec. 4, we mentioned several works which use deterministic world modeling such as David Ha’s original “World Model” paper. We will accordingly refine our wording. We also would like to remark that we choose to use latent states in order to achieve more complex exploration strategies.
> > > >
> > > >
> > > > **About sample-based entropy estimation.**   Sorry for the confusion. SAC does not use closed form entropy estimation. It is a typo here and we should express in the way of “where the entropy can be estimated efficiently, e.g. a factorized (squash) Gaussian policy”. The ease of usage lies in the fact that SAC could use unbiased entropy estimation; on the other hand, the main reason that people do not use latent variable policies in MaxEnt RL (and correctly estimate the true entropy) is that it is hard to estimate the entropy of latent variable policies effectively, which is also why our work is important.
> > > >
> > > > **About Proposition 1.**   We refer to the proof in Sec. D.1 for the mathematical validity of Proposition 1. We recognize that “proposition” is sometimes used to refer to hypotheses, but we use this term to refer to proven results. We have made this more clear in the writing. Intuitively speaking, one only need a single latent variable to approximate any continuous distribution within a small Wasserstein distance. Of course, this is a purely theoretical result and lies on the assumption of arbitrary expressiveness for each Gaussian distribution of the two levels. In practice (such as the examples mentioned by the reviewer from generative modeling), a larger latent dimension would obviously help.

---

> ### Author Response · Authors · 2022-11-16
> **Rebuttal reminder**
>
> Dear reviewer,
>
> It has been a while since we post our response. Therefore, we would like to ask if you have any further questions, especially about normalizing flow experiments that have been added to the paper or anything else we may be able to help address your concern.

---

> ### Author Response · Authors · 2022-11-25
> **Looking forward to your response**
>
> Dear reviewer fjCR,
>
> Thanks again for your detailed comments. We hope our responses have addressed your concerns, and thus request for a reconsideration of the score. We would appreciate it if we can get your further feedback at your earliest convenience.

---

> > ### Comment · Reviewer_fjCR · 2022-12-05
> > **Response to authors**
> >
> > Apologies for the delayed response. Thank you for the updates to the text of the paper, as well as the additional experiments. I have increased my score accordingly and have updated my review.

---

> > > ### Author Response · Authors · 2022-12-05
> > > **Thank you for the feedback!**
> > >
> > > We thank you for the reconsideration of score on our work! Glad to see that our explanation and illustration gain your recognition. If you also have any further question by any chance, please feel free to reach out again.

---

### Official Review · Reviewer_cPcG · 2022-10-24

**Confidence:** 3
**Correctness:** 4
**Technical Novelty And Significance:** 2
**Empirical Novelty And Significance:** 2
**Recommendation:** 6

**Clarity, Quality, Novelty And Reproducibility:**

The paper has good clarity, but some details about the algorithms need to be explained more.

The novelty of the proposed algorithm is not enough. The two main techniques in SMAC seem to be borrowed from previous literature, though may not in the RL papers. The factorized Gaussian distribution has been investigated a lot in previous research.


**Strength And Weaknesses:**

Strength:

The paper has good writing and clear delivery of the idea. Two techniques for solving the entropy estimation and variance reduction are effective. Experiments show performance improvement over previous algorithms almost consistently. Necessary ablation studies are conducted.

Weakness:

Unclear about Eq. (11). In Eq. (11), only the first sampling $s^{(0)}$ is used for sampling the action, while the other K $s$ are used for estimating the marginal entropy. Why is it? Why not use all $s$ for sampling action as well as estimating the entropy? Please explain in the paragraph.

Explanations about notations. I’m not clear about the upper scripts $(a), (b)$ in Eq.(12). Please explain this in the paragraph.

Missing references and baselines. There is a related paper (Probabilistic mixture-of-experts for efficient deep reinforcement learning. Jie et al. 2021) which also adopts the Gaussian mixture model for policy function approximation. Also I think Dreamer-v2 should also serve as an important baseline method. The paper should compare the proposed method with these.

The variance reduction in Sec. 3.2.2 seems to be a computationally heavy one. Please compare the computational efficiency in the alation study about whether this variance reduction is applied or not.

How about the on-policy algorithms? The proposed SMAC algorithm seems to only be compared with off-policy algorithms like SAC and TD3. How about on-policy algorithms like PPO? Is SMAC possible to be applied on PPO with an on-policy update as well?

In sec 5.1, is this model-free setting? If so, please write it explicitly in the paragraph.

“Reasonable number of particles”. In Sec.5.1, the paper suggests to use a reasonable number of particles. What specific number is it, 8? What’s the number of the particles in each experiment?

Performance improvement in most of the experiments over existing algorithms seems to be marginal.


**Summary Of The Paper:**

The paper proposes to use the latent variable policy for MaxEnt framework, which is testified to improve the sample efficiency and stability of RL. It uses some techniques to estimate the marginal entropy, as well as reducing the variances in the gradient. Experiments on DM control suite show the performance improvement of the proposed SMAC algorithm.


**Summary Of The Review:**

Overall, the paper is written well. But the novelty of the proposed algorithm is not high enough, and the experiments show marginally improved performance in most environments. Some important baselines are missed, and more experiments should be added to clarify the problems I asked in the Weakness section.

---

> ### Author Response · Authors · 2022-11-10
> **Response to Reviewer cPcG (1)**
>
> Thanks for the detailed review. We hope our response could resolve your concern such that you would like to increase the score. Below we post our response to each of the concerns.
>
> **About Equation 11.**  Constructing a valid lower bound on the entropy of latent variable policies is not trivial, and had been regarded as difficult until some recent works. Our choice of estimator requires a specific design to guarantee its validity, whereas naively taking multiple samples provides an unbounded upper bound, which during training explodes, verified in Fig. 3.  Please refer to Appendix D.2 for the principles behind this estimator. From a high level point of view, one would achieve a variational upper bound of entropy when using an arbitrary variational distribution, while only when conditioning on the true posterior distribution (of latent given action and state) can we obtain an entropy lower bound. We gain access to this true posterior via exploiting the property that $\pi(a|s)q(s|h)=p(s|a, h)\pi(a|h)$, but this only holds for the latent used for sampling the action $a$. Therefore, we can only achieve one sample ($s^{(0)}$) from the true posterior.
>
> **About notations (a)(b) in Equation 12.**  As we mentioned in Sec. 3.2.2., for the $2^l$ samples of every level $l$, we use (a) to denote the entropy estimator calculated with the first $2^{l-1}$ samples, and use (b) to denote the entropy estimator calculated with the second half $2^{l-1}$ samples. This is an antithetic sampling method that allows us to theoretically (and practically) achieve a lower variance than the simple nested estimator.
>
> **Missing references and baselines.**  Thanks for mentioning the work using mixture-of-experts for policy approximation! We have cited it in the revision version. In light of this being brought up, we have also conducted a comparison with a mixture-of-Gaussian baseline following the practice of [1]. We show the experiments in Fig. 11 in Appendix C, together with the comparison with the normalizing flow based method. Although probabilistic MOE methods bring a certain degree of improvements, extensive experiments show that SMAC achieves better sample efficiency on most of the environments. See Appendix C for more discussion.
>
> Furthermore, we would like to point out that the latent variable model has a natural connection with mixture-of-experts. From a theoretical perspective, a latent variable model is an infinite mixture of expert distributions, which is actually the key of the proof of Proposition 1 in Sec. D.1. However, while an explicit mixture model requires predefining the number of components, our use of a nested estimator is only used for entropy estimation. We find that we can easily recover a multimodal mixture behavior; see simple continuous bandits in Sec. B.1.
>
> About DreamerV2, would you like to elaborate more on why you think it should be an important baseline? Our proposed methodology provides an agnostic approach to integrating world models into a latent variable policy, and an efficient entropy estimation method for doing maximum entropy RL. Therefore, our method is orthogonal to DreamerV2. In this work we do not testify SMAC upon the world modeling method of DreamerV2 as it uses a discretized latent space and thus would make use of alternative tricks to become performant, but we agree this is an important and promising future direction.
>
> [1] Jie Ren, et al. Probabilistic Mixture-of-Experts for Efficient Deep Reinforcement Learning.

---

> > ### Author Response · Authors · 2022-11-10
> > **Response to Reviewer cPcG (2)**
> >
> > **Computation efficiency of multi-level Monte Carlo (MLMC) estimator.**   As we have stated in Sec. 3.2.2, although MLMC looks like a complicated variance reduction method, its computation be easily parallelized for every level and particle, and the overall cost becomes negligible compared to main computational overhead such as convolution computation in feature extraction. As a result, MLMC brings almost no additional computational expense. To further demonstrate this point, we report the speed of nested estimator and MLMC estimator in the setup of Sec. 5.2, both with $32$ particle samples. The average frame per second (FPS) on first ten episodes for nested estimator and MLMC are $33.45$ and $33.89$, respectively. This shows MLMC only brings $1.315%$ more computation in training cost.
> >
> > **About on-policy algorithms.**  We agree that PPO is an important model-free RL baseline with wide application. However, we do not present the result of PPO in Sec. 5.1 since in mojuco-based environments, PPO is usually considered to have worse sample efficiency compared to off-policy methods like SAC [1, 2], especially when the interaction with environment is expensive. Further, extending PPO to use a latent variable policy requires accurate estimation of likelihood ratios, which is a different non-trivial problem but could definitely be worth pursuing. Ultimately, PPO is not a maximum entropy RL algorithm, thus we do not investigate more on this aspect as it is not very related to our methodology.
> >
> > **About Sec. 5.1.**  We are very sorry for the confusion. Yes, Sec. 5.1 is in a model-free setup. We point out this in the final version.
> >
> > **About number of particles.**   We are very sorry about the missing description. For the number of particles, we simply tried random values in {8, 16, 32} (this is because MLMC requires the number of samples to be a power of two), with different results depending on the difficulty of the environment. See our ablation experiment for analysis.
> >
> > **About the improvement.**
> > As shown in Fig. 4, 6, 10 and 11, our method achieves better sample efficiency in the majority of cases among the twenty-ish environments from DeepMind Control Suite. We want to strengthen that even on the fully observed environments where our method shares similar final performance with baseline, the proposed algorithm can more reliably find optimal policies with lower variance and is thus more appealing. This is purely an exploration problem, so any policy will converge when given sufficient training time. Being more reliable showcases the ability to better track multiple optimal trajectories during the training phase. With noisy observations, our approach also outperforms in terms of final performance; one can see that in Table 1, in many cases the std intervals do not overlap.
> >
> > [1] Tuomas Haarnoj, et al. Soft Actor-Critic: Off-Policy Maximum Entropy Deep Reinforcement Learning with a Stochastic Actor
> > [2] Kei Ota, et al. Can Increasing Input Dimensionality Improve Deep Reinforcement Learning?

---

> ### Author Response · Authors · 2022-11-16
> **Rebuttal reminder**
>
> Dear reviewer,
>
> It has been a while since we post our response. Therefore, we would like to ask if you have any further questions, especially about normalizing flow experiments, other algorithmic details, or anything else we may be able to help address your concern.

---

> ### Author Response · Authors · 2022-11-25
> **Looking forward to your response**
>
> Dear reviewer cPcG,
>
> Thanks again for your detailed comments. We hope our responses have addressed your concerns, and thus request for a reconsideration of the score. We would appreciate it if we can get your further feedback at your earliest convenience.

---

> ### Author Response · Authors · 2022-12-07
> **Request for an update of reviews for our submission**
>
> Dear Reviewer cPcG,
>
> We thank for your effort for reviewing our paper. We have demonstrated the universal benefit of our method and the improvement upon other methods. Since the deadline of discussion period is about to end, and other reviewers have given their feedbacks, we are very expectant to see your re-evaluation of our latest replies.
>
> Thank you very much!

---

> ### Author Response · Authors · 2022-12-09
> **Do our responses clarify your concerns?**
>
> Dear Reviewer cPcG,
>
> We thank you again for your insightful review! We appreciate the points you raise regarding the comparison with other methods. We have addressed these issues in our updated draft. Your feedback has helped us make the paper much clearer. We would appreciate it if you could take the updated draft into consideration in your evaluation of the paper. We are also happy to answer any further questions you might have before the end of the discussion period!

---

> ### Author Response · Authors · 2022-12-11
> **Looking forward to your feedback (at the last day)**
>
> Dear Reviewer cPcG,
>
> We hope it does not disturb you. Once again, we appreciate your thoughtful and enlightening remarks. We would appreciate additional feedback from you, as there is only 1 day remaining before the rebuttal phase concludes. We believe our responses have allayed your concerns. Based on your and the other reviewers' suggestions, we think the manuscript now meets higher standards. If necessary, we would be happy to have any additional discussions.
>
> Best regards,
> Authors

---

> > ### Comment · Reviewer_cPcG · 2022-12-12
> > **Response to rebuttal**
> >
> > Sorry for the late reply. After reading the rebuttal, I decide to raise the score.

---

> > > ### Author Response · Authors · 2022-12-12
> > > **Thanks for the reconsideration.**
> > >
> > > We thank you for the reconsideration of our submission! Happy to see that our explanation and illustration gain your recognition. If you also have any further question by any chance, please feel free to reach out again.

---

### Official Review · Reviewer_y62x · 2022-10-25

**Confidence:** 3
**Correctness:** 4
**Technical Novelty And Significance:** 2
**Empirical Novelty And Significance:** 3
**Recommendation:** 6

**Clarity, Quality, Novelty And Reproducibility:**

The writing is clear, and the contribution is novel to the best of my knowledge. I would believe that the results are reproducible.


**Strength And Weaknesses:**

Strengths
The general idea makes sense, and the methodology is well explained. The technical contribution is not novel, but applying it to the RL setting appears to be novel.

Weaknesses
The experiments don't seem to compare against energy based model and autoregressive model variants. The introduction says these "complicate training procedures and are inefficient in practice", but it seems important to provide experimental evidence.


**Summary Of The Paper:**

This paper considers the problem of MaxEnt reinforcement learning, with the goal of increasing the expressiveness of the policy while still allowing for practical entropy maximization. The proposed method relies on latent variables to avoid complexities with using EBMs.

The challenge with directly using latent variables is that the entropy is difficult to estimate, therefore entropy maximization is difficult. Typically, we'd want to optimize latent variable models via variational approaches with the ELBO, but this gives a lower bound of likelihood, hence an upper bound of the entropy term and the maxent rl objective. Maximizing the upper bound of course leads to practical issues.

Instead the paper proposes to use a technique in hierarchical inference (Sobolev & Vetrov 2019), which gives a lower bound estimate to the entropy term. The rough intuition is use multiple samples, but unlike IWAE, do so in a way that reuses samples, and this can be shown to flip the direction of the bound.

With this new estimator, the paper optimizes the maxent rl objective with variational methods on a latent variable model and shows improvements over SAC.


**Summary Of The Review:**

In summary the paper proposes a method for optimizing latent variable policies with the maxent RL objective, which is difficult to do with naive techniques.

---

> ### Author Response · Authors · 2022-11-10
> **Response to Reviewer y62x**
>
> Thank you for your insightful comments. We hope that we could resolve your concern such that we could receive a better score. Below we provide responses to the questions.
>
> **About comparison with other probabilistic modeling methods.**  We acknowledge that it is important to compare with other generative modeling methods, and we have added comparison with normalizing flow-based and mixture-of-expert based policies in Fig. 11. The empirical comparison shows that the proposed SMAC generally enjoys a better sample efficiency. On the other hand, we have not found previous work using autoregressive models in policy modeling. This is probably due to their difficulty in usage in the sense of efficient sampling: sampling from an autoregressive model is extremely slow and can result in multiple times' slowdown during training. The [1] mentioned in Section 1 models the dynamics model rather than the policy in an autoregressive way. Regarding energy-based modeling (EBM), [2] proposes soft-Q learning with the help of energy modeling. However, due to the usage of SVGD, which does the inference in the kernelized functional space, soft-Q learning is computationally expensive and unstable, plus its performance in high dimensional action space is worse than SAC (in fact, it is a precursor of SAC). Therefore, we do not empirically compare soft-Q learning and the proposed SMAC method.
>
> We are happy to answer any other questions you may have.
>
> [1] Michael Zhang, et al. Autoregressive dynamics models for offline policy evaluation and optimization.
> [2] Tuomas Haarnoj, et al. Reinforcement Learning with Deep Energy-Based Policies.

---

### Author Response · Authors · 2022-11-10
**General Response (pdf updated)**

We thank all reviewers for their feedback and for taking the time to discuss our work. We agree with many points that were brought up which have led us to improve our work. We have revised the writing of our submission, improved the wording to avoid confusion, and included additional experiments to demonstrate and ablate the effectiveness of the proposed methodology. Please see our revision, where we have highlighted modifications in blue to ease reading. We briefly summarize points that were brought up by multiple reviewers:

**Comparison to alternative stochastic policies.** We have included additional empirical comparisons with normalizing flow and mixture-of-experts based policies. These can be found in Figure 11. Our approach retains the best sample efficiency, requires fewer parameters and compute, and does not require tuning of architecture. Detailed comparisons and summary of analyses are in the individual responses for each reviewer.

**Using more samples naively is not sufficient.** We emphasize that naively using more samples results in an entropy upper bound, optimization of which can lead to catastrophic failure. Our method constructs a lower bound on the entropy by using samples from the true posterior. However, simply using more particles again does not improve performance and actually results in a similar training curve as SAC (see our ablation experiment). This is because using more samples naively creates a worse signal-to-noise ratio, resulting in variance dominating when the number of samples increases. The advantage of SMAC is a joint effect from multiple different aspects, starting from principled observations and resulting in a simple and low-cost approach to more flexible stochastic policies.

We have responded to every comment individually. We hope to see your individual responses for additional in-depth discussions.

---

### Decision · Program_Chairs · 2023-01-20

**Decision:**

Accept: poster

**Justification For Why Not Higher Score:**

* The idea is not transformative.  There are alternative techniques based on mixture models and normalizing flows, but it is not clear whether the latent variable model has an inherent theoretical advantage.
* The proposed method requires the estimation of an upper bound on entropy that is conceptually complex and requires multi-level samping.
* The empirical results show a slight edge for the proposed technique, but it is not clear that the improvement is significant


**Justification For Why Not Lower Score:**

* The problem of modeling multimodality in max entropy RL is an important one
* The use of a latent variable model to achieve multimodality in stochastic policies is novel
* The empirical results demonstrate the effectiveness of the approach

The reviewers all agreed that this paper should be published and will be disappointed if it is not published.

**Metareview: Summary, Strengths And Weaknesses:**

The paper describes a new max entropy RL technique that enables multimodal stochastic policies via a latent variable model.

Strengths:
* The problem of modeling multimodality in max entropy RL is an important one
* The use of a latent variable model to achieve multimodality in stochastic policies is novel
* The empirical results demonstrate the effectiveness of the approach

Weaknesses:
* The idea is not transformative.  There are alternative techniques based on mixture models and normalizing flows, but it is not clear whether the latent variable model has an inherent theoretical advantage.
* The proposed method requires the estimation of an upper bound on entropy that is conceptually complex and requires multi-level samping.
* The empirical results show a slight edge for the proposed technique, but it is not clear that the improvement is significant

Since the weaknesses are not significant and the proposed latent variable policies will be of interest to the RL community, this work deserves to be published.

**Note From Pc:**

if the above contains the word "oral" or "spotlight" please see: "oral" presentation means -> notable-top-5% and "spotlight" means -> notable-top-25%. As stated in our emails, we are disassociating presentation type from AC recommendations

**Summary Of Ac-Reviewer Meeting:**

The discussion focused on the following strengths and weaknesses:

Strengths:
* The problem of modeling multimodality in max entropy RL is an important one
* The use of a latent variable model to achieve multimodality in stochastic policies is novel
* The empirical results demonstrate the effectiveness of the approach

Weaknesses:
* The idea is not transformative.  There are alternative techniques based on mixture models and normalizing flows, but it is not clear whether the latent variable model has an inherent theoretical advantage.
* The proposed method requires the estimation of an upper bound on entropy that is conceptually complex and requires multi-level samping.
* The empirical results show a slight edge for the proposed technique, but it is not clear that the improvement is significant

The reviewers all agreed that this paper should be published.  While we have a consensus, the scores remain borderline because the work is not groundbreaking, but still interesting for the RL community.